# Conformal Prediction for Time-series Forecasting with Change Points

**Sophia Sun, Rose Yu**
Computer Science and Engineering
University of California, San Diego

## Abstract

Conformal prediction has been explored as a general and efficient way to provide uncertainty quantification for time series. However, current methods struggle to handle time series data with change points — sudden shifts in the underlying data-generating process. In this paper, we propose a novel Conformal Prediction for Time-series with Change points (CPTC) algorithm, addressing this gap by integrating a model to predict the underlying state with online conformal prediction to model uncertainties in non-stationary time series. We prove CPTC's validity and improved adaptivity in the time series setting under minimum assumptions, and demonstrate CPTC's practical effectiveness on 6 synthetic and real-world datasets, showing improved validity and adaptivity compared to state-of-the-art baselines. Our code is available at `https://github.com/Rose-STL-Lab/CPTC`.

## 1 Introduction

Uncertainty Quantification (UQ) is a key building block of reliable machine learning systems. Conformal prediction (CP) is a popular distribution-free UQ method that provides finite sample coverage guarantees without placing assumptions on the underlying data generating process [50, 32]. Because of its simplicity and generality, conformal prediction has seen wide adoption in many tasks, ranging from healthcare [8], robotics [34], to validating the factuality of generative models [17].

Existing CP algorithms for time series forecasting mostly adopt an online learning framework [24, 4], where the prediction interval adapts to distribution changes in data *reactively*. The online CP algorithms achieve a asymptotic marginal coverage guarantee (the miscoverage rate converges to a specified fraction as the time horizon goes to infinity). As a result, the algorithms inevitably exhibit miscoverage when distribution shifts occur, and then have to over- or under-cover in later timesteps to compensate. This behavior is evident in the ACI and CP baselines in Figure 1. This is undesirable in practice as these under-covered periods may incur a lot of risk.

This paper builds upon the observation that in real-world scenarios, distribution shifts in time-series are often *predictable*. Take for example the task of forecasting electricity demands: we know that the hidden dynamics may differ between day and night, or during weekdays and weekends. The same logic applies to traffic forecasting and product demand forecasting, where there are patterns in the shifts between "surge times" and "normal times". In this work, we study time series data that exhibit this type of abrupt shifts, or change points, in the underlying generative process.

State Space Models (SSMs) [28, 10] are a powerful tool for modeling time series data, as they provide a structured framework to capture the underlying temporal dependencies through latent states. In particular, Switching Dynamical Systems (SDS) [1] is a type of SSM with an additional set of latent variables (known as *switches*) that represent the operating mode active at the current timestep. SDS introduces the inductive biases that allow SSMs to switch between a discrete set of dynamics, which can be learned from training data. It is shown to be a flexible, robust, and interpretable representation of time series with varying dynamics exhibiting change points [9, 16]. SDS is deployed not only for

39th Conference on Neural Information Processing Systems (NeurIPS 2025).

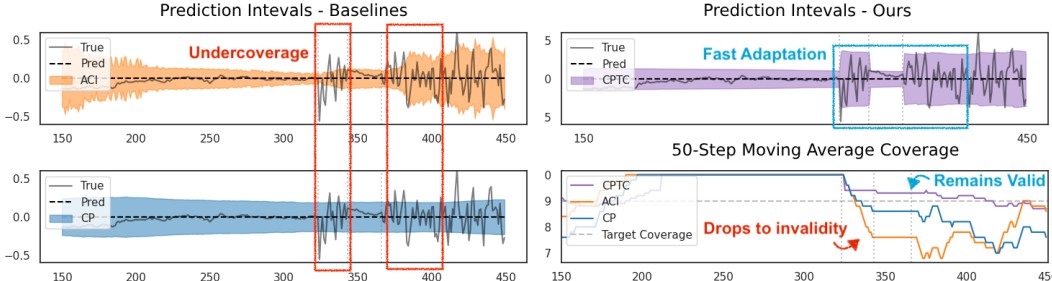

Figure 1: Comparison of the prediction intervals obtained by our algorithm CPTC (purple) against online conformal prediction baselines on synthetic data. The vertical dashed line marks the distribution shifts; ideal behavior is consistent coverage at the horizontal dashed line in the final panel. Bottom right panel shows that CPTC achieves fast adaptation and remains valid when change points occur.

energy [29, 43, 20] and traffic [59, 15] forecasting as examples given above, but also on a wide range of applications including neuroscience [25, 41, 30], engineering [45, 53], and sports analytics [58], all would benefit from robust and calibrated uncertainty quantification.

Our contributions are:

- We propose a new algorithm **C**onformal **P**rediction for **T**ime series with **C**hange points (CPTC) that utilizes state transition predictions to improve uncertainty quantification for time-series forecasts. Leveraging properties of a SDS model, CPTC consolidates multiple future forecasts to adaptively adjust its prediction intervals when underlying dynamics shift.

- We prove that CPTC achieves asymptotic valid coverage, *without assumptions* on the data generation process or state transition model accuracy. When predicted state transitions align well with distribution shifts, CPTC can anticipate uncertainty and adapt faster.

- We show strong empirical results on 3 synthetic and 3 real-world datasets. Compared to state-of-the-art CP baselines, CPTC achieves more robust coverage with comparable prediction intervals sharpness (example in Figure 1), and is computationally light.

## 2 Related Work

**Probabilistic Forecasting for Time-Series with Change Points.**   Probabilistic forecasting has become central to modern time-series analysis by modeling a distribution over future outcomes rather than a single-point forecast [13]. Classic approaches often adopt Bayesian or approximate Bayesian strategies for uncertainty quantification: for instance, Bayesian Neural Networks (BNNs) [52, 54, 21] or neural process models [22, 46] use neural networks to parameterize the underlying stochastic process. Other methods, such as DeepAR [42], PatchTST [39], or Temporal Fusion Transformer [33], generate probabilistic forecasts directly by minimizing a pinball loss (i.e., quantile loss), thereby learning predictive distributions.

A key challenge arises when *change points* occur - i.e., abrupt distribution shifts in the time series [38]. Change point detection and segmentation have been studied extensively (beyond our scope, see [3] for a recent survey). Modeling time series with change points typically features state-space models (SSM), and in particular switching dynamical systems (SDS) [27, 9, 37]. These SDS approaches use variational inference to fit neural parameterizations of transitions and emission distributions. However, existing probabilistic forecasters often lack strict calibration guarantees and can be overconfident in practice [31], and many require specialized architecture tuning for different tasks, limiting their applicability to real-world UQ challenges.

**Conformal Prediction for time-series.**   Conformal prediction (CP) [50] has emerged as a leading framework for UQ due to its algorithmic simplicity, broad applicability, and finite-sample coverage guarantees; see [7] for a comprehensive introduction. For time-series forecasting, some works focus on joint coverage over fixed-length horizons [44, 47, 62]; others assume stable underlying dynamics and adapt only to shifts in residual distributions [56] . [48] uses known covariate shifts (by propensity weighting) to achieve theoretical coverage, but can fail in settings where the shift mechanism is unknown, which is typical for time series forecasting. [26] introduces a multi-model ensemble CP framework, optimizing model selection within a pool of experts for combined coverage. CP has also been extended for change-point detection in time series via conformal martingales [51, 49].

Recent works on online conformal prediction develop online strategies that guarantee marginal coverage for adversarial or nonstationary data streams. ACI [23], AgACI [60], DtACI [24], and Conformal PID control [4] leverage online optimization and have a "reaction" period when distribution shifts. Multivalid Prediction (MVP) [12] provides group coverage guarantees and learns a calibration threshold online, although it can require longer calibration windows. Closer to our work, SPCI [57] learns an autocorrelative estimation of the residual's quantiles through Quantile Random Forest; whereas HopCPT [11] condition their conformal interval on similar parts of the time series using a Modern Hopfield Network. The drawback of these two methods is that the regression models are trained during inference time, which (1) requires a long cold start window to learn meaningful associations and (2) is computationally expensive. Our contribution is a CP algorithm that is computationally light during inference time, and can leverage state prediction capabilities within the SDS model to anticipate and adapt quickly to distribution changes.

## 3 Background

### 3.1 Conformal Prediction

We briefly review the algorithm and guarantees for conformal prediction, and refer readers to [7] for a thorough introduction. In this paper CP refers to *split conformal prediction* (we consider full conformal prediction out of scope, because its computationally complexity renders it nonviable for deep learning settings). The goal of conformal prediction is to produce a *valid* prediction interval (Def. 3.1) for any underlying prediction model.

**Definition 3.1** (Validity). Given a new data point $(x, y)$, where $y$ is the prediction target and $x$ is the covariates, and a desired confidence $1 - \alpha \in (0, 1)$, the prediction interval $\Gamma^{1-\alpha}(x)$ is a subset of $\mathcal{Y}$ containing probable outputs given $x$. The region $\Gamma^{1-\alpha}$ is valid if

$$P(y \in \Gamma^{1-\alpha}(x)) \geq 1 - \alpha \tag{1}$$

Let $\mathcal{D} = \{(x^i, y^i)\}_{i=1}^n$ be a dataset whose data points $(x^i, y^i)$ are sampled from a distribution on $\mathcal{X} \times \mathcal{Y}$. Conformal prediction splits the dataset into a proper training set $\mathcal{D}_{train}$ and a calibration set $\mathcal{D}_{cal}$. A prediction model $\hat{f} : \mathcal{X} \to \mathcal{Y}$ is trained on $\mathcal{D}_{train}$. We use a *nonconformity score* function $A : \mathcal{X} \times \mathcal{Y} \to \mathbb{R}_{\geq 0}$ to quantify how well a data sample from calibration *conforms* to the training dataset. Typically, we choose a metric of disagreement between the prediction and the true label as the non-conformity score, such as the Euclidean distance:

$$A(x, y) \overset{\text{e.g.}}{=} d(y, \hat{f}(x)) \overset{\text{e.g.}}{=} \|y - \hat{f}(x)\|_2 \tag{2}$$

Let $\mathcal{S} = \{A(x^i, y^i)\}_{(x^i, y^i) \in \mathcal{D}_{cal}}$ denote the set of nonconformity scores of all samples in the calibration set $\mathcal{D}_{cal}$. During inference time given a new data $x^{n+1}$, the conformal prediction interval is constructed as in Eqn 3, where $Q$ is the empirical quantile function.

$$\Gamma^{1-\alpha}(x^{n+1}) := \{y : A(x^{n+1}, y) \leq Q^{1-\alpha}(\mathcal{S} \cup \{\infty\})\} \tag{3}$$

We say a sample is *covered* if the true value lies in the prediction interval $y^{n+1} \in \Gamma^{1-\alpha}(x^{n+1})$ Conformal prediction is guaranteed to produce valid prediction intervals [50] if the calibration data and test data are exchangeable (in a dataset $\{(x^i, y^i)\}_{i=1}^n$ of size $n$, any of its $n!$ permutations are equally probable).

Conformal prediction has been extended to the online setting. More precisely, the algorithm observes $(x_1, y_1), (x_2, y_2), \ldots$ sequentially, and needs to construct a prediction set $\Gamma_t^{1-\alpha}$ for $y_t$ at time step $t$. Without making any distributional assumptions on the data generation process, online CP algorithms achieve asymptotic guarantee as Eqn 4, where $\mathbb{1}\{\cdot\}$ is the indicator function:

$$\lim_{T \to \infty} \frac{1}{T} \sum_{t=1}^{T} \mathbb{1}\{y_t \notin \Gamma_t^{1-\alpha}(x_t)\} = \alpha \tag{4}$$

We refer readers to [6] for a summary of theoretical results in this setting.

### 3.2 Switching Dynamical Systems for Modeling Time-series with Change Points

Switching dynamics systems (SDS), or mixed dynamics systems, provide a powerful framework for modeling time series with change points, where the underlying dynamics shift between distinct

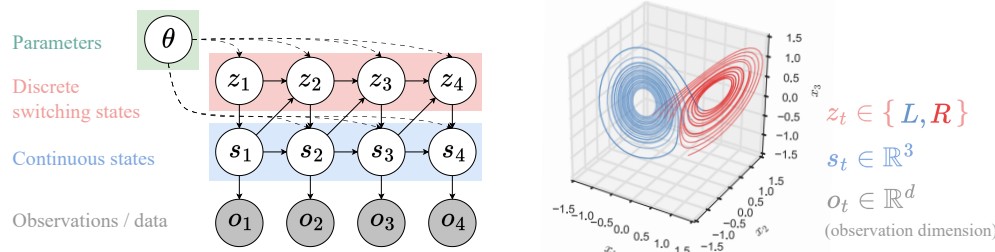

Figure 2: (Left) Generative model of the SDS dynamics as in Eqn 5. Shaded circles represent observed variables; hollow circles are latent variables. (Right) Example to illustrate notation. We show a Lorenz attractor, a canonical nonlinear dynamical system, approximated by a linear SDS [35].

regimes. SDS assumes the system consists of a total of $K$ different base dynamics. It explicitly incorporates discrete switching *states* to represent the abrupt shifts among base dynamics. We denote the discrete switching states at timestep $t$ as $z_t \in \mathcal{Z}$, $\mathcal{Z} = \{1, \ldots, K\} \subset \mathbb{Z}^+$, to index one of $K$ base dynamical systems, and represent the observed sequence as $o_1, \ldots, o_T$ or $o_{1:T}$ for conciseness.

There have been many different variations and implementations of SDS. Classic SDS have *memoryless* state transitions, i.e. $p(z_t|z_{t-1})$ is parameterized by a stochastic transition matrix $\mathbf{A} \in \mathbb{R}^{K \times K}$. This results in an "open loop" system and the state duration follows a geometric distribution. Recent works have sought to fix this issue, for example by closing the loop using recurrent dependencies on $s_t$ [36], or modeling explicit durations [9]. Our generative model is illustrated in figure 2): for generality, we factor out any non-Markovian covariates to model parameters $\theta$. The joint distribution can be factorized as

$$P(o_{1:T}, s_{1:T}, z_{1:T}) = \prod_{t=1}^{T} P(o_t|s_t)P(s_t|s_{t-1}, z_t, \theta)P(z_t|s_{t-1}, z_{t-1}, \theta) \tag{5}$$

where $p(s_1|z_1)p(z_1)$ is the initial state prior. The $K$ dynamical systems have continuous state transition $p(z_t|z_{t-1}, s_{t-1}, \theta)$, and the emission $p(o_t|s_t)$ can be linear or nonlinear (e.g. parameterized by a neural network). Moreover, the SDS framework naturally supports forecasting both the next observation $o_{t+1}$ via $P(o_{t+1} \mid s_{t+1})$ and the next switching state $z_{t+1}$ via $P(z_{t+1} \mid s_t, z_t, \theta)$.

## 4 Conformal Prediction for Time series with Change points (CPTC)

We introduce CPTC, a novel conformal prediction algorithm that utilizes the state prediction from SDS to improve uncertainty quantification for time series forecasting.

**Problem Setting**  We are given a sequence of observations $o_t \in \mathbb{R}^d$ for all $t$ potentially with change points. To be consistent with the standard notation in conformal prediction literature, let $\mathcal{X} \subseteq \mathbb{R}^m$ denote the input feature domain and $\mathcal{Y} \subseteq \mathbb{R}^d$ the target observation space. $\mathcal{Z} = \{1, \ldots, K\} \subset \mathbb{Z}^+$ indexes one of $K$ underlying states. The input features can include past observations and other environmental covariates. In our implementations, $m = k \cdot d$ with $k$ being the look back window.

We consider the online conformal prediction setting where we have a sequence of data points from $\mathcal{X} \times \mathcal{Y}$: $(x_0, y_0), (x_1, y_1), \ldots, (x_T, y_T)$. We have a trained state predictor for $\hat{p}(z_t|x_{0:t})$ where $z_t \in \mathcal{Z}$ is the true (latent switching) state at time $t$, a state-conditioned forecaster $\hat{f} : \mathcal{X} \times \mathcal{Z} \to \mathcal{Y}$ (a probabilistic SDS model can serve both purposes), and are given a desired confidence level $1 - \alpha$. At each time step $t$, we have seen $(x_1, y_1), (x_2, y_2), \ldots, (x_{t-1}, y_{t-1})$ and the next input feature vector $x_t$. We want to produce an adaptive, time-dependent prediction interval $\Gamma_t(x_t) \subseteq \mathcal{Y}$ for the unseen true target $y_t$, which we will learn at the next time step. Our goal is to produce prediction intervals with correct empirical coverage, i.e.

$$P(y_t \in \Gamma_t(x_t)) \geq 1 - \alpha \ \text{ for any } t \geq 0 \tag{6}$$

### 4.1 Algorithm

Existing online conformal prediction algorithms often rely on assumptions about how distribution would shift in time series. This is infeasible for data with change points due to unknown shift mechanism. The main contribution our algorithm makes towards addressing this issue is *anticipating* this type of abrupt shifts. Our CPTC algorithm is outlined in pseudo-code in algorithm 1. In this section, we discuss the general procedure and key components of the algorithm.

---

**Algorithm 1:** Conformal Prediction for Time series with Change points (CPTC)

**Input:** nonconformity score function $A$, probabilistic state model $\hat{p}(z_t|x_{0:t})$ with prior $\hat{p}(z_0)$,
forecaster model $\hat{f} : \mathcal{X} \times \mathcal{Z} \rightarrow \mathcal{Y}$, confidence level $1 - \alpha$, learning rate $\gamma$

**Output:** Prediction intervals $\Gamma(x_t)$ for $t \geq 1$

**1** **for** $z \in \mathcal{Z}$ **do**
**2** $\quad$ Initialize $\mathcal{S}_z \leftarrow \{\}$, $\alpha_{z,0} \leftarrow \alpha$
**3** **end for**
**4** **if** *exists warm start data* $\{(x_t, y_t)\}_{t=-w}^{0}$ **then**
**5** $\quad$ **for** $t \in [-w, 0]$ **do**
**6** $\quad\quad$ $\mathcal{S}_{z_t} \leftarrow \mathcal{S}_{z_t} \cup \{A(x_t, y_t)\}$, where $z_t \sim \hat{p}(z_t = z|x_{0:t})$; // `warm start scores`
**7** $\quad$ **end for**
**8** **end if**
**9** **for** $t \in [1, T]$ **do**
**10** $\quad$ **for** $z \in \mathcal{Z}$ **do**
**11** $\quad\quad$ $\Gamma_{z,t}(x_t) \leftarrow \{y : A(x_t, y) \leq Q^{1-\alpha_{z,t}}(\mathcal{S}_z \cup \{\infty\})\}$; // `state-specific CP`
**12** $\quad$ **end for**
**13** $\quad$ $\Gamma_t(x_t) \leftarrow$ Aggregate the $\Gamma_{z,t}(x_t)$s by Eqn 10 or Eqn 11;
$\quad\quad$ **Output:** $\Gamma_t(x_t)$
**14** $\quad$ Sample state $\hat{z}_t \sim \hat{p}(z_t|x_{0:t})$; // `state-specific coverage target tracking`
**15** $\quad$ Update $\alpha_{\hat{z}_t, t+1} \leftarrow \alpha_{\hat{z}_t, t} + \gamma \cdot (\alpha - err_t)$, where $err_t = \mathbb{1}\{y_t \notin \Gamma_t(x_t)\}$;
**16** $\quad$ Update scores $\mathcal{S}_{\hat{z}_t} \leftarrow \mathcal{S}_{\hat{z}_t} \cup \{A(x_t, y_t)\}$;
**17** **end for**

---

**Switching Behavior and SDS Integration.** We explicitly model the abrupt shift in dynamics with the switching dynamical systems (SDS) to anticipate the change. In our formulation, the latent state $z_t$ indicates which dynamical regime (out of $K$ possible regimes) is currently active at time $t$. Given the SDS formulation (e.g. Eqn 5), we can factor the state transition probability from the conformal prediction process and do calibration for each of the $K$ dynamics separately. The goal in Eqn 6 can therefore be written as:

$$\sum_{z \in \mathcal{Z}} P(y_t \in \Gamma_t(x_t)|z_t = z) \, P(z_t = z|x_{0:t}) \geq 1 - \alpha \tag{7}$$

**State Prediction.** The SDS model provides us with the state predictor $\hat{p}(z_t \mid x_t)$. In practice, SDS models [9, 36] have existing $z_t$ forecasts that we can extract. For other model architectures, one can modify them to output an extra covariate prediction for the state, or add an auxiliary model to classify or cluster the input space into different regimes. In practice, $z_t$ doesn't necessarily have to come from a model, and can be any discrete contextual variable such as weekday/weekend/holiday or day/night.

**Initialization and Warm Start,** For each switching state $z \in Z$, we initialize (1) a mode-specific set of nonconformity scores $\mathcal{S}_z$ and (2) the mode-specific confidence level $\alpha_{z,0} = \alpha$. If there are $w$ steps of warm-start data $\{(x_t, y_t)\}_{t=-w}^{0}$, we initialize the sore sets $\mathcal{S}_z$ by sampling $z_t \sim \hat{p}(z_t = z|x_{0:t})$ for $t = -w, \dots, 0$, and inserting $A(x_t, y_t)$ into $\mathcal{S}_{z_t}$. Depending on the size of the warm-start window and desired properties, practitioners can adjust the warm-starting approach. For example, using the entire warm-start window $\mathcal{S}_z = \{A(x_t, y_t)\}_{t=-w}^{0}$ for all $z \in \mathcal{Z}$ will result in better stability but less adaptability to modes. The approach does not change the algorithm's coverage guarantees.

**State-Specific Prediction Intervals.** At every time step $t \in [1, T]$, CPTC creates state-specific conformal prediction intervals for all states with nonzero probability. This allows us to generate predictions tailored to the current anticipated dynamics. Specifically, for every state $z \in \mathcal{Z}$ where $\hat{p}(z|x_{0:t}) > 0$, we first obtain the point prediction through forecaster model $\hat{f}(x_t, z)$, and then the state-specific prediction interval $\Gamma_{z,t}(x_t)$ by calibrating on nonconformity scores set $\mathcal{S}_z$ to the adaptive confidence level of $1 - \alpha_{z,t}$.

**Online Conformal Prediction.** The exchangeability assumption does not apply to the online setting of our work. If we naively calibrate with online data, when the data distribution changes, the coverage probability may deviate from the target level $1 - \alpha$. Therefore, we follow online CP methods such as ACI [23] to address this problem by continuously updating $\alpha_{z,t}$ to track a surrogate miscoverage rate $\alpha_{z,t}^*$ as defined in eqn 8 (for each state $z$ in our setting), an internal estimate of the target coverage.

$$\alpha_{z,t}^* = \sup\{\beta \in [0,1] : M_{z,t}(\beta) \leq \alpha\}, \quad M_{z,t}(\alpha) = p(A(x_t, y_t) > Q^{1-\alpha}(\mathcal{S}_z)) \tag{8}$$

In eqn 8 $M_{z,t}$ measures the probability that the true label $y_t$ falls outside the prediction set $\Gamma_{z,t}$ when the predicted state is $z$. We track $\alpha^*_{z,t}$ over time by updating the estimate $\alpha_{z,t}$ with online optimization. In our implementation, we use the simple update from ACI (Eqn 9), though more sophisticated approaches such as [4] may be used in its place without affecting the overall algorithm.

$$\alpha_{\hat{z}_t,t+1} \leftarrow \alpha_{\hat{z}_t,t} + \gamma \cdot (\alpha - err_t), \text{ where } err_t = \mathbb{1}\{y_t \notin \Gamma_t(x_t)\} \tag{9}$$

Here $\gamma > 0$ is the step size hyperparameter, and $\hat{z}_t \in \mathcal{Z}$, $\hat{z}_t \sim \hat{p}(z_t = z | x_{0:t})$ is the state at time $t$ sampled to be updated. The update rule increases $\alpha_{\hat{z}_t,t}$ if coverage is too conservative and decreases it otherwise, ensuring that for each $z \in \mathcal{Z}$, $\alpha_{z,t}$ converges to a value that maintains long-term coverage.

**Aggregation.** We aggregate the prediction intervals for each state $\Gamma_{z,t}(x_t)$ into one final set $\Gamma_t(x_t)$ using the weighted level set, as illustrated in Eqn 10.

$$\Gamma_t(x_t) := \left\{ y : \sum_{z \in \mathcal{Z}} \hat{p}(z_t = z \mid x_t) \mathbb{1}\{ y \in \Gamma_{z,t}(x_t)\} \geq 1 - \alpha \right\} \tag{10}$$

Solution to the constraint in Eqn 10 is the minimal set that achieves the marginal coverage. It can be obtained through discretizing $\mathcal{Y}$ into a fine grid and calculating probability mass at each point, which becomes computationally expensive as the grid resolution or dimensionality increases. For faster computation and in implementation, we approximate the weighted level-set in Eqn 10 by taking the union of intervals of the most probable states as in Eqn 11. The two aggregation strategies achieve similar results on all our datasets (Appendix B.4).

$$\Gamma_t(x_t) \approx \cup_{z \in \mathcal{Z}'_t} \Gamma_{z,t}(x_t), \text{ where } \mathcal{Z}'_t = \underset{S \subseteq \mathcal{Z}}{\arg\min} |S| \text{ s.t. } \sum_{z \in S} p(z_t = z \mid x_t) \geq 1 - \alpha. \tag{11}$$

**Modularity and the role of the state model.** Our algorithm can be viewed as running multiple instances of online conformal inference (lines 14-16 of algorithm 1), one for each underlying state, and adaptively aggregating the confidence intervals when the underlying mode is switching. This differs from the setting where the temporal correlation between residuals needs to be learned online [24, 12, 57, 11]; the state model allows us to leverage training data for such information. Our algorithm is modular - the state model, forecaster model, and the online adaptive conformal prediction algorithms all operate independently of each other. For example, the state-specific online coverage tracking (line 15 of algorithm 1) can be replaced with other online conformal prediction variants for data-specific applications. Our theoretical guarantees are agnostic to the forecaster models used.

## 4.2 Theoretical analysis

In this section, we discuss theoretical guarantees of the CPTC algorithm. In particular, we establish finite-sample validity under exchangeability, asymptotic coverage in dynamic (potentially nonstationary) settings, robustness to imperfect state classification, and faster adaptation to distribution shifts that align with model-predicted state changes. See Appendix A for proofs and theoretical details.

**Finite-sample validity under exchangeability.** We start by showing that when data is exchangeable (e.g. independent and identically distributed), CPTC achieves finite sample validity. The proof is standard based on showing the marginal coverage of split conformal prediction.

**Proposition 4.1** (Finite-sample validity under exchangeability). *If the data* $(x_t, y_t)$, $t \geq 1$ *are exchangeable, prediction intervals obtained via Algorithm 1 satisfy* $P(y_t \in \Gamma_t(x_t)) \geq 1 - \alpha$.

**Asymptotic validity without assumptions.** Theorem 4.2 ensures that CPTC provides reliable uncertainty quantification in the long run *without assumptions on time-series stationarity or accurate state predictions*. (We do assume that the distribution of states and state predictions exhibit stable long-term behavior in Assumption 1.) We achieve our desiderata of Eqn 6 on average as $T$ grows asymptotically, consistent with the results of other online conformal prediction works.

**Theorem 4.2** (Asymptotic validity of CPTC). *For any sample size* $T \geq 1$, *the CPTC algorithm (with weighted average aggregation) achieves:*

$$\left| \frac{1}{T} \sum_{t=1}^{T} \mathbb{E}[err_t] - \alpha \right| \leq \frac{1}{T} \sum_{z=1}^{K} \frac{1 - (1 - c\gamma)^{|\mathcal{T}_z|}}{\gamma} |\alpha - \alpha^*_z|$$

*Where:*

- $err_t = \mathbb{1}[Y_t \notin \Gamma_t(x_t)]$, *the miscoverage rate.*
- $\mathcal{T}_z = \{t \in \{1, \ldots, T\} : \hat{z}_t = z\}$, *the set of time steps the predicted state is $z$.*
- $\alpha_z^*$ *is the optimal miscoverage target for timesteps $\mathcal{T}_z$.*
- $c$ *is a miscoverage constant in Lemma A.1 [23].*

*Note the RHS of Eqn 4.2 decays at a rate of $\mathcal{O}(T)$ and satisfies $\lim_{T \to \infty} \left| \frac{1}{T} \sum_{t=1}^{T} \mathbb{E}[err_t] - \alpha \right| = 0$.*

**Robustness to imperfect state prediction.** In real-world applications, the state predictor $\hat{p}(z_t | x_{0:t})$ that plays a large role in our algorithm is often imperfect. The CPTC algorithm accommodates these scenarios by tracking $\alpha_z^*$ for each *predicted* mode $z$ with adaptive online updates. Theorem 4.3 quantifies the effect of imperfect state prediction by showing that the miscoverage rate is bounded by the product of the misclassification rate and state-specific deviations.

**Theorem 4.3** (Finite-Sample Miscoverage Bound with Imperfect State Predictions). *For any sample size $T \geq 1$, the CPTC algorithm ensures that:*

$$\left| \frac{1}{T} \sum_{t=1}^{T} \mathbb{E}[err_t] - \alpha \right| \leq \epsilon \cdot \max_z \delta_{z,T}$$

*Where $\epsilon = P(\hat{z}_t \neq z_t)$ is the error rate of the state predictions and $\delta_{z,T}$ is the miscoverage deviation from $\alpha$ for any predicted state $z$ defined in Lemma A.1. For all $z$, $\delta_{z,T} \to 0$ as $|T| \to \infty$.*

Theorem 4.3 demonstrates that CPTC can maintain asymptotic valid coverage even when state predictions are imperfect, with the bound tightening as state predictions improve or the number of observations per state increases. In practice, improving the accuracy of the state prediction model directly enhances the coverage performance of CPTC. However, even with imperfect state predictions, the algorithm's adaptive calibration mechanism ensures that the overall coverage remains close to the desired level $1 - \alpha$ as experiments show in Section 5.2.

**Why CPTC? Faster convergence under distribution shifts.** The strength of the CPTC algorithm is its fast adaptation in dynamic environments with distributional changes. When a distribution shift aligns with a predicted state transition, the structure of CPTC allows the confidence level $\alpha_{z,t}$ to converge to the new target error rate faster compared to purely online methods.

Consider a data stream with a distribution shift occurring at time $t_{\text{shift}}$. Let $\alpha_j^*$ denote the optimal target error rate for mode $j$ after the shift, and assume that the predicted state $\hat{z}_t$ correctly reflects the shift: $t_{\text{shift}}$ and $\alpha_z^* = \alpha_j^*$ for $t > t_{\text{shift}}$. For conciseness of notation in Theorem 4.4, let $\delta_{j,T}$ and $\delta_{ACI,T}$ denote the miscoverage rate *starting from* $t_{\text{shift}}$ for CPTC and ACI respectively.

**Theorem 4.4** (Miscoverage Ratio under State-Coincident Distribution Shift). *Under a state shift from $i$ to $j$ at time step $t_{shift}$, coinciding with predicted transition, the CPTC algorithm achieves faster convergence to the new target $\alpha_j^*$ compared to non-state-aware algorithm ACI at a ratio of:*

$$\frac{\delta_{j,T}}{\delta_{ACI,T}} \leq \frac{|\alpha_{j,t_{shift}-1} - \alpha_j^*|}{|\alpha_{t_{shift}-1} - \alpha_j^*|}$$

The numerator $|\alpha_{j,t_{\text{shift}}-1} - \alpha_j^*| \to 0$ as $T \to \infty$, when in-state adaptation has converged; but the denominator $|\alpha_{t_{\text{shift}}-1} - \alpha_j^*|$ remains greater than zero regardless of the length of the time series. This result highlights the advantage of segmenting nonconformity scores by state: the state-specific $\alpha_{z,t}$ aligns the target miscoverage updates with distribution shifts, allowing CPTC to quickly adjust its prediction intervals. Such accelerated adaptation allows for shorter miscoverage periods and is critical in real-world applications where rapid responses to changing conditions are necessary.

## 5 Experiments

**Baselines.** We selected the following baseline methods. **RED-SDS** (Recurrent Explicit Duration Switching Dynamical System) [9] is a state-of-the-art Bayesian model that learns state transitions (implementation details in Appendix B.1). We also use REDSDS as the base predictor for mode switching for our method in the experiments. **CP** is a generalization of conformal prediction to the online setting, also known as Online Sequential Split Conformal Prediction in [60]. We choose **ACI**

[23] to represent online CP algorithms leveraging online optimization. **SPCI** [57] and **HopCPT** [11] are state-of-the-art CP algorithms that learns adaptive predictive intervals by quantile regression and Modern Hopfield Networks learned from the time series respectively.

**Metrics.** We evaluate calibration and sharpness for each method. For *calibration*, we report the empirical coverage on the test set. Coverage should be as close to the desired confidence level $1 - \alpha$ as possible. $\text{Coverage} = \frac{1}{T} \sum_{t=1}^{T} \mathbb{1}(y_t \in \Gamma_t(x_t))$

For *sharpness*, we report the average width or area of the Prediction Intervals (PI). The measure should be as small as possible while being valid (coverage maintains above the specified confidence level). $\text{Width} = \frac{1}{T} \sum_{t=1}^{T} |\Gamma_t(x_t)|$

**Datasets.** The three synthetic datasets are designed with increasing randomness in mode changes, challenging the adaptivity of CPTC. **Bouncing Ball** is comprised of univariate time series encoding the height of a ball bouncing between two walls with constant velocity and elastic collisions, following [18, 9]. The two switching states are going up/down, each associated with a different level of Gaussian noise added to observation (Bouncing Ball obs), or the underlying dynamics (Bouncing Ball dyn) which induces uncertainty in the phase as well. **3-mode system** is a switching linear dynamical system with 3 switching states, where each mode samples from a Poisson distribution for duration.

For real-world datasets, the **Electricity** and **Traffic** datasets from [19] have hourly frequency and exhibit seasonality both in terms of the time series itself and volatility. The **honey bee trajectory** dataset [40] is the most complex, composed of 4-dimensional trajectories with length averaging to 900 frames, where the bees' dance can be decomposed into "left turn", "right turn" and "waggle".

Illustrations of the datasets and detailed description can be found in Appendix B. The warm-start window for synthetic and real datasets are 50 and 100 time steps respectively (excluding bees, whose $w = 15$). The datasets represent a diverse set of scenarios to demonstrate the robustness and versatility of CPTC, capturing the complexity and variability of practical time-series forecasting problems.

### 5.1 Analysis of Coverage and Sharpness Results

Results for uncertainty quantification in table 4 show that we achieve significantly better validity on all datasets compared to baselines. RED-SDS's intervals fail due to lack of calibration, a common pitfall of probabilistic methods (shown also in [57]). CP's under-coverage shows that the data are not exchangeable. In the presence of change points, ACI's coverage fluctuates dramatically and did not converging within the horizon (more visualizations in Appendix B.5). For SPCI and HopCPT, which also guarantees asymptotic validity, the invalidity is likely because they are designed for large datasets ($T \geq 10,000$ for HopCPT), and did not have enough data to react or learn useful residual patterns for the relatively short prediction horizon of our datasets ($T = 200$ for synthetic datasets, 300 for electricity and traffic, and 60 for bees). Figure 3 provides a qualitative example of how the prediction intervals compare across different methods. Our method (purple shaded regions) does not over-cover during nonvolatile hours, nor under-cover during busy ones, as does ACI.

Table 1: Performance on synthetic and real-world datasets with target confidence $1 - \alpha = 0.9$ (for horizon $T = 200$ for synthetic datasets, 300 for electricity and traffic, and 60 for bee, mean $\pm$ standard deviation of the test samples). Methods that are *invalid* (coverage below 90%) are grayed out. Our method achieves a high level of calibration (coverage is close to 90%) consistently.

| | | | RED-SDS | CP | ACI | SPCI | HopCPT | Ours |
|---|---|---|---|---|---|---|---|---|
| Bouncing Ball obs. | Cov | | 16.90 $\pm$ 12.96 | 87.67 $\pm$ 6.54 | 89.82 $\pm$ 2.95 | 87.13 $\pm$ 4.79 | 79.45 $\pm$ 12.51 | 90.15 $\pm$ 1.19 |
| | Width | | 1.60 $\pm$ 0.07 | 10.54 $\pm$ 3.19 | 3.85 $\pm$ 2.60 | 2.82 $\pm$ 0.41 | 1.98 $\pm$ 0.90 | 3.71 $\pm$ 0.98 |
| Bouncing Ball dyn. | Cov | | 13.20 $\pm$ 11.50 | 86.45 $\pm$ 12.28 | 89.38 $\pm$ 2.56 | 89.16 $\pm$ 3.47 | 81.12 $\pm$ 7.85 | 90.47 $\pm$ 2.30 |
| | Width | | 1.37 $\pm$ 0.05 | 11.49 $\pm$ 4.39 | 2.27 $\pm$ 1.65 | 2.26 $\pm$ 0.28 | 2.95 $\pm$ 1.12 | 1.76 $\pm$ 0.71 |
| 3-Mode System | Cov | | 93.02 $\pm$ 3.04 | 63.98 $\pm$ 5.04 | 89.90 $\pm$ 2.25 | 84.90 $\pm$ 5.41 | 90.85 $\pm$ 5.90 | 94.96 $\pm$ 1.96 |
| | Width | | 2.08 $\pm$ 0.16 | 1.46 $\pm$ 0.50 | 6.40 $\pm$ 1.92 | 3.70 $\pm$ 0.89 | 9.07 $\pm$ 1.23 | 2.45 $\pm$ 0.72 |
| Traffic | Cov | | 23.15 $\pm$ 10.95 | 87.06 $\pm$ 3.71 | 90.01 $\pm$ 0.87 | 84.38 $\pm$ 2.47 | 88.91 $\pm$ 3.55 | 92.38 $\pm$ 1.24 |
| | Width | | 0.05 $\pm$ 0.02 | 21.91 $\pm$ 10.42 | 27.81 $\pm$ 54.08 | 5.32 $\pm$ 1.95 | 7.50 $\pm$ 10.09 | 7.92 $\pm$ 2.98 |
| Electricity | Cov | | 62.85 $\pm$ 13.67 | 85.69 $\pm$ 6.17 | 89.79 $\pm$ 0.83 | 84.10 $\pm$ 2.77 | 86.50 $\pm$ 2.66 | 91.22 $\pm$ 1.29 |
| | Width | | 162.67 $\pm$ 811.73 | 366.05 $\pm$ 2280.45 | 45.74 $\pm$ 279.21 | 228.14 $\pm$ 1207.58 | 155.71 $\pm$ 130.43 | 139.75 $\pm$ 620.44 |
| Dancing Bees | Cov | | 84.92 $\pm$ 6.84 | 79.86 $\pm$ 20.73 | 86.25 $\pm$ 8.10 | 79.20 $\pm$ 0.30 | 72.11 $\pm$ 1.84 | 92.64 $\pm$ 3.19 |
| | Width | | 0.25 $\pm$ 0.02 | 1.65 $\pm$ 0.58 | 4.79 $\pm$ 4.02 | 1.77 $\pm$ 1.38 | 1.06 $\pm$ 0.51 | 0.79 $\pm$ 0.27 |

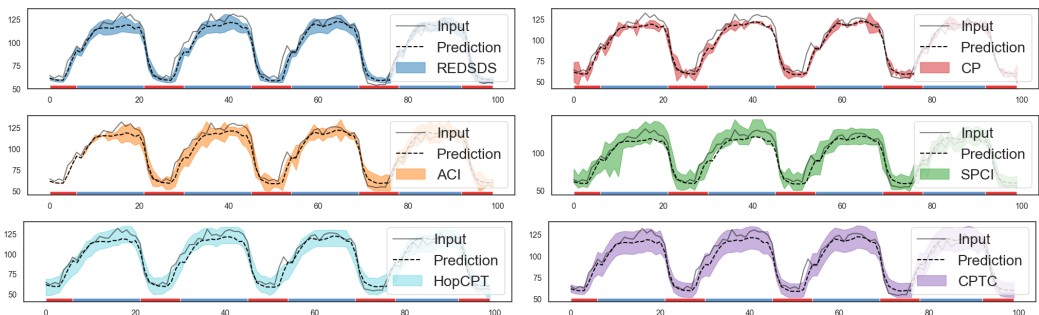

Figure 3: **Visualization of prediction intervals on the Electricity hourly demand dataset.** The red and blue bars in the bottom reflects the underlying switching state of day and night. Our method (purple) adapts to different levels of volatility between day and night, and achieves stabler coverage over time, whereas ACI (yellow) over-covers during the night and under-covers at change points.

## 5.2 Ablation Studies

**Robustness to $p(\hat{z})$ accuracy.**    We conduct ablation studies on the 3 synthetic datasets of which we have ground truth underlying states. As shown in table 2, it is evident that increasing the error in the state prediction does not affect the coverage performance. This verifies the robustness of our algorithm and theoretical results (Theorem 4.2 and 4.3). Table 3 further shows that the error of state prediction reflects on the width of the prediction intervals: while remaining the same level of coverage, the better the state predictions, the sharper the intervals.

Table 2: Coverage of CPTC for using ground truth (GT) labels and with various levels of injected noise, on synthetic data with $T = 200$. We can see that coverage performance does not change significantly with error in state prediction.

Table 3: Interval width under the same setting. While the coverage guarantee is the same, the more accurate the state prediction is, the sharper the intervals.

|  | Ground truth state labels | GT w/ 20% error | GT w/ 50% error |
|---|---|---|---|
| BB obs. | 90.31 $\pm$ 1.27 | 90.27 $\pm$ 1.30 | 90.13 $\pm$ 1.40 |
| BB dyn. | 90.34 $\pm$ 0.93 | 90.32 $\pm$ 0.96 | 90.33 $\pm$ 1.05 |
| 3-mode | 92.67 $\pm$ 1.52 | 92.70 $\pm$ 1.63 | 92.53 $\pm$ 1.54 |

|  | GT | w/ 20% err | w/ 50% err |
|---|---|---|---|
| BB obs. | 3.61 $\pm$ 0.93 | 3.75 $\pm$ 0.99 | 3.94 $\pm$ 1.17 |
| BB dyn. | 2.09 $\pm$ 0.60 | 2.15 $\pm$ 0.62 | 2.18 $\pm$ 0.65 |
| 3-mode | 2.04 $\pm$ 0.67 | 2.19 $\pm$ 0.76 | 2.27 $\pm$ 0.89 |

Additional ablation studies are presented in Appendix B.4. On **Aggregation Methods** (by discretization as Eqn 10 or union as Eqn 11), we found that on our benchmark datasets, union is a close proximation of the true objective, and CPTC achieves similar width and coverage on a sampled subset. For **Long Horizon Forecasting** where $T \geq 10,000$, our method still achieves valid and stable coverage, but have wider PI width compared to SPCI an HopCPT. They achieve this by frequently re-training their models online, which CPTC does not and is less computationally demanding.

## 6    Conclusion and Discussion

In this paper, we introduced Conformal Prediction for Time-series with Change points (CPTC), a novel conformal prediction algorithm for uncertainty quantification in non-stationary time series. By leveraging state information in switching dynamics systems models, CPTC offers improvements in adaptivity over existing conformal prediction methods. Our theoretical guarantees ensure validity *without assumptions* on time-series stationarity or accurate state predictions. Empirical results corroborate our theory and demonstrate the effectiveness of CPTC across diverse synthetic and real-world datasets, achieving robust coverage under distributional shifts with comparable sharpness compared to state-of-the-art baselines. The adaptivity advantage is most pronounced in shorter time series, when baselines require more data to react or learn temporal correlations.

Limitations of CPTC include (1) slower convergence rate in scenarios with frequent or unpredictable state transitions, as the algorithm requires sufficient observations within each state to achieve optimal calibration and sharpness, (2) wider prediction interval width when applied to very long time series compared methods specialized for such scenarios (e.g. HopCPT), and (3) requirement of a state prediction model, making our method primarily applicable to SDS models and discrete SSMs. Future work could explore extending the framework to continuous states and the theoretical implications thereof, and studying the decision theoretic properties of using conformal prediction for safety-critical applications requiring real-time adaptation.

## Acknowledgments and Disclosure of Funding

This work was supported in part by the U.S. Army Research Office under Army-ECASE award W911NF-07-R-0003-03, the U.S. Department Of Energy, Office of Science, IARPA HAYSTAC Program, and NSF Grants #2205093, #2146343, #2134274, CDC-RFA-FT-23-0069, DARPA AIE FoundSci and DARPA YFA.

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

# A Theoretical Results and Proofs

## A.1 Setup and Notation

Let $z_t \in \mathcal{Z} = \{1, \ldots, K\}$ denote the unobserved discrete mode, $x_t \in \mathcal{X}$ denote the continuous state, and $y_t \in \mathcal{Y}$ the observation at time $t$. We assume we have access to a probabilistic model $\hat{p}(z_t|X_{1:t-1})$ that estimates the mode $\hat{z}_t$.

For a target coverage rate $1 - \alpha$, let $\Gamma_t(X_t)$ be the prediction set at time $t$. We define the miscoverage indicator $M_t$ as follows, and aim to bound the overall miscoverage rate over $T$ time steps:

$$M_t = \frac{1}{T} \sum_{t=1}^{T} err_t, \quad \text{where} \quad err_t := \mathbb{1}\big(Y_t \notin \Gamma_t(X_t)\big),$$

## A.2 Proofs: Lemma

In practical scenarios, the state prediction model may not be perfectly accurate. We now analyze the impact of imperfect state predictions on the coverage guarantees of the CPTC algorithm.

We derive the following lemma from the convergence result of Proposition 4.1 in [23] with two additional assumptions also according to [23]: (1) the existence of a single fixed optimal target $a^* \in [0,1]$, and (2) $\mathbb{E}[err_t|a_t] = M(a_t)$. These are reasonable assumptions that allow us to quantify distribution shifts and are necessary for analyzing adaptation behavior (whereas proposition 4.1 itself is a worst-case upper bound.) We refer the readers to the original paper for the complete derivation.

**Lemma A.1** (Adaptive Conformal Inference Miscoverage Bound Within Predicted States). *Let $\mathcal{T}_z = \{t \in \{1, \ldots, T\} : \hat{z}_t = z\}$ denote the set of times when the predicted state is $z$. For any $T > 1$ and predicted state $\hat{z}_t = z$, the Adaptive Conformal Inference (ACI) algorithm ensures that there exists a constant $\delta_{z,T}$ such that:*

$$\left| \frac{1}{|\mathcal{T}_z|} \sum_{t \in \mathcal{T}_z} \mathbb{E}[err_t] - \alpha \right| \leq \delta_{z,T}$$

*Explicitly,*

$$\delta_{z,T} = \frac{1 - (1 - c\gamma)^{|\mathcal{T}_z|}}{|\mathcal{T}_z|\gamma} |\alpha - \alpha_z^*|$$

*where*

- $\alpha_z^*$ *is the optimal miscoverage target for timesteps $\mathcal{T}_z$.*
- $\gamma$ *is the step size*
- $c$ *the miscoverage constant.*

*Note that $\delta_{z,T}$ decays at a rate of $\mathcal{O}(T)$ and satisfies $\lim_{\mathcal{T}_z \to \infty} \delta_{z,T} = 0$.*

## A.3 Proof of Theorem 4.2

**Theorem** ( 4.2 Asymptotic validity of CPTC ). For any sample size $T \geq 1$, without placing any assumptions, the CPTC algorithm ensures that:

$$\left| \frac{1}{T} \sum_{t=1}^{T} \mathbb{E}[err_t] - \alpha \right| \leq \frac{1}{T} \sum_{z=1}^{K} |\mathcal{T}_z|\delta_{z,T} = \frac{1}{T} \sum_{z=1}^{K} \frac{1 - (1 - c\gamma)^{|\mathcal{T}_z|}}{\gamma} |\alpha - \alpha_z^*|$$

which decays at a rate of $\mathcal{O}(T)$ and satisfies

$$\lim_{T \to \infty} \left| \frac{1}{T} \sum_{t=1}^{T} \mathbb{E}[err_t] - \alpha \right| = 0.$$

To prove theorem 4.2, we assume that both the distribution of the underlying modes $z$ and the predictions $\hat{z}$ is stationary (Assumption 1).

**Assumption 1** (Stationary Distribution of States). The sequence of true states $\{z_t\}$ has a stationary distribution $\pi(z)$, and the sequence of predicted states $\{\hat{z}_t\}$ also has a stationary distribution $\hat{\pi}(z)$. Specifically:

$$\forall z, \in \mathcal{Z}, \quad \lim_{t \to \infty} p(z_t = z) = \pi(z), \quad \lim_{t \to \infty} p(\hat{z}_t = z) = \hat{\pi}(z),$$

This assumption is well-justified in practical scenarios where the sequence of observations and predictions arises from processes that, while non-stationary over short intervals, exhibit stable long-term behavior. In time-series or dynamic environments, regularities in data generation or prediction models often lead to empirically observed stationarity in distributions. While restrictive, this assumption aligns with prior work in sequential and online prediction frameworks, where stationarity assumptions are standard to derive theoretical guarantees.

*Proof of Theorem 4.2.* Let $\mathcal{T}_z = \{t \in \{1, \ldots, T\} : \hat{z}_t = z\}$ represent the set of timesteps when the predicted state is $z$. For the total number of timesteps $T$, the overall miscoverage error can be expressed as:

$$\frac{1}{T} \sum_{t=1}^{T} \mathbb{E}[err_t] = \frac{1}{T} \sum_{z=1}^{K} \sum_{t \in \mathcal{T}_z} \mathbb{E}[err_t]$$

$$= \frac{1}{T} \sum_{z=1}^{K} |\mathcal{T}_z| \left( \frac{1}{|\mathcal{T}_z|} \sum_{t \in \mathcal{T}_z} \mathbb{E}[err_t] \right).$$

From Lemma A.1, for any predicted state $z$, the miscoverage within $\mathcal{T}_z$ satisfies:

$$\left| \frac{1}{|\mathcal{T}_z|} \sum_{t \in \mathcal{T}_z} \mathbb{E}[err_t] - \alpha \right| \leq \delta_{z,T},$$

where:

$$\delta_{z,T} = \frac{1 - (1 - c\gamma)^{|\mathcal{T}_z|}}{|\mathcal{T}_z|\gamma} |\alpha - \alpha_z^*|.$$

Substituting this bound into the overall miscoverage error, we obtain:

$$\left| \frac{1}{T} \sum_{t=1}^{T} \mathbb{E}[err_t] - \alpha \right| \leq \frac{1}{T} \sum_{z=1}^{K} |\mathcal{T}_z| \delta_{z,T}.$$

Expanding $\delta_{z,T}$, the overall bound becomes:

$$\left| \frac{1}{T} \sum_{t=1}^{T} \mathbb{E}[err_t] - \alpha \right| \leq \frac{1}{T} \sum_{z=1}^{K} \frac{1 - (1 - c\gamma)^{|\mathcal{T}_z|}}{\gamma} |\alpha - \alpha_z^*|.$$

By assumption 1, as $T \to \infty$, the size of $|\mathcal{T}_z|$ for each state $z$ grows proportionally to $T$. The term $1 - (1 - c\gamma)^{|\mathcal{T}_z|}$ approaches 1, while $|\mathcal{T}_z|/T$ converges to the relative proportion of time spent in state $z$, denoted as $p_z$. Hence, as $T \to \infty$, the overall bound $\left| \frac{1}{T} \sum_{t=1}^{T} \mathbb{E}[err_t] - \alpha \right| \to 0$.

Thus, the CPTC algorithm ensures that the overall error converges to the target coverage level $\alpha$ at a rate of $\mathcal{O}(T)$, satisfying:

$$\lim_{T \to \infty} \left| \frac{1}{T} \sum_{t=1}^{T} \mathbb{E}[err_t] - \alpha \right| = 0.$$

$\square$

## A.4 Proof of Theorem 4.3

**Theorem** ( 4.3 Finite-Sample Miscoverage Bound with Imperfect State Predictions)**.** For any sample size $T \geq 1$, the CPTC algorithm ensures that:

$$\left| \frac{1}{T} \sum_{t=1}^{T} \mathbb{E}[err_t] - \alpha \right| \leq \epsilon \cdot \max_z \delta_{z,T}$$

where:

- $err_t := \mathbb{1}\big(Y_t \notin \Gamma_t(X_t)\big)$, the miscoverage rate.

- $\mathcal{T}_z = \{t \in \{1, \dots, T\} : \hat{z}_t = z\}$, the set of times when the predicted state is $z$.

- $\epsilon = \mathbb{P}(\hat{z}_t \neq z_t)$ is the misclassification rate of the state predictions.

- $\delta_{z,T}$ is the deviation from $\alpha$ within any predicted state $z$ as in Lemma A.1.

*Proof of Theorem 4.3.* First we will partition time into correct vs. incorrect predictions. Define:

$$\mathcal{C} = \{t : \hat{z}_t = z_t\}, \quad \mathcal{I} = \{t : \hat{z}_t \neq z_t\}, \quad \text{and} \quad \frac{|\mathcal{I}|}{T} = \epsilon.$$

**Calibrated expected coverage on correctly predicted states** $\mathcal{C}$**.** Since there are no distribution shifts within each state, we have the same expected error from calibration data. Therefore,

$$\mathbb{E}[err_t] = \alpha \implies \mathbb{E}[err_t] - \alpha = 0.$$

**Bounded deviation on incorrectly predicted states** $\mathcal{I}$**.** On $\mathcal{I}$, we have $\big|\mathbb{E}[err_t] - \alpha\big| \leq \delta_{z,T}$. Therefore,

$$\sum_{t=1}^{T} \big|\mathbb{E}[err_t] - \alpha\big| = \sum_{t \in \mathcal{C}} \big|\mathbb{E}[err_t] - \alpha\big| + \sum_{t \in \mathcal{I}} \big|\mathbb{E}[err_t] - \alpha\big| \leq 0 + |\mathcal{I}| \, \delta_{z,T} = |\mathcal{I}| \, \delta_{z,T}.$$

Since $\frac{|\mathcal{I}|}{T} = \epsilon$, dividing by $T$ yields

$$\frac{1}{T} \sum_{t=1}^{T} \big|\mathbb{E}[err_t] - \alpha\big| \leq \epsilon \, \delta_{z,T}.$$

**Combine to achieve the overall bound.** By the triangle inequality, we know that

$$\left| \frac{1}{T} \sum_{t=1}^{T} \mathbb{E}[err_t] - \alpha \right| \leq \frac{1}{T} \sum_{t=1}^{T} \big|\mathbb{E}[err_t] - \alpha\big|.$$

Hence, from the previous step,

$$\left| \frac{1}{T} \sum_{t=1}^{T} \mathbb{E}[err_t] - \alpha \right| \leq \epsilon \, \delta_{z,T}.$$

Since on $\mathcal{I}$ the predicted state $\hat{z}_t$ could vary among different $z$ values, we take $\delta_{z,T} \leq \max_z \delta_{z,T}$. Thus,

$$\left| \frac{1}{T} \sum_{t=1}^{T} \mathbb{E}[err_t] - \alpha \right| \leq \epsilon \max_z \delta_{z,T}.$$

$\square$

## A.5  Corollaries

Theorem 4.3 indicates that the overall miscoverage rate deviates from the desired level $\alpha$ by at most $\epsilon \cdot \max_z \delta_{z,T}$. The misclassifications rate $\epsilon$ contributes directly to the deviation, and the calibration error $\delta_{z,T}$ within each predicted state decreases as more data becomes available. As $T$ increases and $\max_z \delta_{z,T} \to 0$, the deviation is primarily governed by the misclassification rate $\epsilon$.

**Corollary A.2** (Zero miscoverage rate under Accurate State Prediction). *If predicted state probabilities are accurate $\hat{p}(z_t|x_{1:t-1}) = p(z_t|x_{1:t-1})$, then $\epsilon = 0$, therefore $\mathbb{E}[err_t] = \alpha$ for all $T$, achieving optimal performance of online conformal prediction.*

**Corollary A.3** (Asymptotic calibration under eventually correct State Prediction). *If the predicted state probabilities $\hat{p}(z_t|x_{1:t-1})$ converge in probability to the true state distribution $p(z_t|x_{1:t-1})$ as $T \to \infty$. With $T \to 0$, $\epsilon \to 0$ as well. Hence, the overall miscoverage rate converges asymptotically to 0.*

## A.6  Proof of Theorem 4.4

Consider a data stream with a distribution shift occurring at time $t_{\text{shift}}$. Let $\alpha_j^*$ denote the optimal target error rate for mode $j$ after the shift, and assume that the predicted state $\hat{z}_t$ correctly reflects the shift: $t_{\text{shift}}$ and $\alpha_z^* = \alpha_j^*$ for $t > t_{\text{shift}}$. For conciseness of notation in Theorem 4.4, let $\delta_{j,T}$ and $\delta_{ACI,T}$ denote the miscoverage rate *starting from* $t_{\text{shift}}$ for CPTC and ACI respectively.

**Theorem** (4.4 Miscoverage Ratio under State-Coincident Distribution Shift). Under a state shift from $i$ to $j$ at time step $t_{\text{shift}}$, coinciding with predicted transition, the CPTC algorithm achieves faster convergence to the new target $\alpha_j^*$ compared to non-state-aware algorithm ACI at a ratio of:

$$\frac{\delta_{j,T}}{\delta_{\text{ACI},T}} \leq \frac{|\alpha_{j,t_{\text{shift}}-1} - \alpha_j^*|}{|\alpha_{t_{\text{shift}}-1} - \alpha_j^*|}$$

*Proof of Theorem 4.4.* We want to compare the convergence behavior of CPTC and a non-segmented online conformal prediction algorithm (e.g., ACI) when a distribution shift occurs at time $t_{\text{shift}}$ and this shift coincides with a predicted state transition from $i$ to $j$.

We assume that State prediction coincident shift, meaning $\alpha_z^* = \alpha_i^*$ for $t \leq t_{\text{shift}}$ and $\alpha_z^* = \alpha_j^*$ for $t > t_{\text{shift}}$.

From Lemma A.1 and Theorem 4.2, we have bounds on these deviations. Taking the ratio of the bounds, we have:

$$\frac{\delta_{j,T}}{\delta_{\text{ACI},T}} \leq \frac{c \cdot \frac{1}{\gamma}|\alpha_{j,t_{\text{shift}}-1} - \alpha_j^*|}{c \cdot \frac{1}{\gamma}|\alpha_{t_{\text{shift}}-1} - \alpha_j^*|} = \frac{|\alpha_{j,t_{\text{shift}}-1} - \alpha_j^*|}{|\alpha_{t_{\text{shift}}-1} - \alpha_j^*|}$$

$\square$

We elaborate on the implication of this result. The ratio indicates that the CPTC achieves lower miscoverage rate when the initial target $\alpha_{j,t_{\text{shift}}-1}$ for state $j$ in CPTC is closer to the new optimal target $\alpha_j^*$ than the global target $\alpha_{t_{\text{shift}}-1}$ used by ACI. This is because CPTC maintains separate targets for each state, allowing it to better track state-specific optimal targets. The base model takes around 2 hours of training time on the GPU

# B  Experiment Details and additional results

## B.1  Hyperparameters for the base forecasting model

We train our RED-SDS on the synthetic datasets, and use the model checkpoints provided by the authors for the real-world datasets. The model architecture consists of a discrete switching component with $K \in \{2, 3\}$ categories and a continuous state space with dimensionality $d_x \in \{2, 4\}$. For training, we employ the ELBOv2 objective with a learning rate $\eta \in [5 \times 10^{-3}, 7 \times 10^{-3}]$, warmup steps of 1000, and gradient clipping at 10.0. The model uses a batch size $B \in \{32, 50\}$ and is trained for $T \in \{20,000, 30,000\}$ steps. The continuous transition and emission models are parameterized by nonlinear MLPs with hidden dimensions $h = 32$, while the inference network uses either a bidirectional RNN or transformer with embedding dimensions $d_e = 4$. For real-world datasets, we apply target transformation and Jacobian correction, while synthetic datasets use raw observations. The model's capacity is controlled through weight decay $\lambda = 10^{-5}$ and MLP hidden dimensions $h \in \{8, 32, 64\}$ depending on the component. For forecasting, the model is trained to forecast a window of $t = 50$ time steps for synthetic datasets (bouncing ball) and a windows of for real-world datasets (electricity, traffic) as specified by their respective metadata. The model generates $N = 100$ $(x_t, y_t, z_t)_{t=0}^{k}$ triplet trajectories from the trained model by Monte Carlo sampling, and use the samples to calculate prediction quantiles for table 1, in accordance to the original paper.s. We segment all datasets by a 70/10/20 train/validation/test split. Conforaml prediction results are reported on the test set only.

## B.2  Computational Resources

All experiments are done on a server machine with an Nvidia A100 GPU, with some data processing and analytics performed on a Apple Macbook Pro laptop computer with M1 chip. As running time vary greatly depending on implementation and hardware, we provide a rough range of algorithm runtime. Training the RED-SDS base model (1 seed) takes 2-5 hours for our datasets, and inference take around 15-20 minutes for all datasets. Among the conformal prediction baselines, SPCI is the most computationally demanding, taking 8-10 hours for inference on our benchmarks. HopCPT follows, taking around 1 hour. CP, ACI, and our method SPCI do not require extensive inference-time computation, completing all inference within 5 minutes.

## B.3  Dataset descriptions, continued

All datasets are visualized in Figure 5 with their underlying states color coded.

**Electricity and Traffic.** The Electricity and Traffic dataset are from the UCI machine learning repository [19] and we use the train/test split from GluonTS [2]. The electricity dataset consists of hourly electricity consumption data for 370 customers, each with 5,833 observations. The traffic dataset consists of hourly occupancy rates from 963 road sensors, each with 4,000 data points.

**Dancing Bees.** The honey bee trajectory dataset is from [40]. Honey bees convey information about the location and distance to a food source through a dance carried out inside the hive. This dance consists of three distinct phases: "left turn", "right turn", and "waggle". The duration and orientation during the waggle phase represent the distance and direction to the food source. Experiments are conducted on 6 trajectories, with lengths of 1058, 1125, 1054, 757, 609, and 814 frames, respectively. In this paper, the prediction interval is on the first two dimensions (the coordinates of the bee) only, but all dimensions are used for prediction.

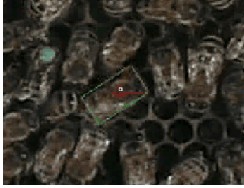 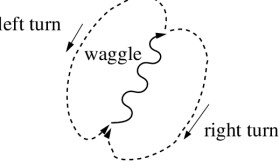

Figure 4: Dancer bees are tracked by a appearance based tracker from video sequences. The tracked bee is shown in green rectangle in the left figure above. The right figure shows a stylized bee dance through which bees talk to the other bees about the orientation and distance to the food sources.

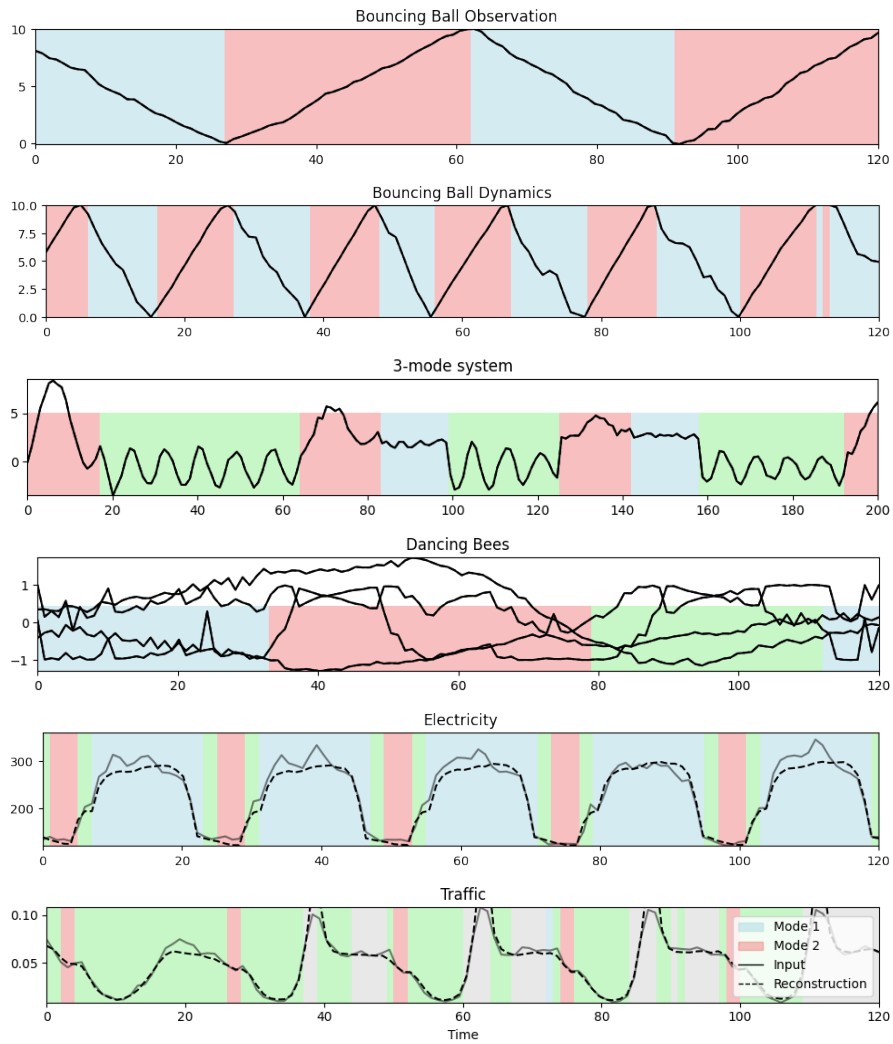

Figure 5: Examples of datasets and prediction results. Dotted line represents prediction whereas solid lines represents ground truth; background colors represent the different operating modes segmented by our base predictor RED-SDS.

## B.4 Additional Experiments

**Additional Baselines**   We add comparison to three more baselines that handles distribution shift in time series data. They are DtACI [5], ECI [55], and L-ARC [61].

We would like to emphasize that these baselines are reactive. The question we try to address with CPTC is - what happens when we can anticipate distribution shift? CPTC is designed such that it greatly improves coverage compared to purely reactive methods, while still maintaining the ability to adapt to unknown shifts. To our knowledge, there are no other models that explore the same problem; CPTC provides practitioners with a simple and sound solution, when they have the ability (such as via a state space model) to predict shifts.

Table 4: Performance on synthetic and real-world datasets with target confidence $1 - \alpha = 0.9$ (for horizon $T = 200$ for synthetic datasets, 300 for electricity and traffic, and 60 for bee, mean $\pm$ standard deviation of the test samples). Methods that are *invalid* (coverage below $90\%$) are grayed out. Our method achieves a high level of calibration (coverage is close to $90\%$) consistently.

| | | DtACI | ECI-cutoff | L-ARC | Ours |
|---|---|---|---|---|---|
| Bouncing Ball obs. | Cov | 92.45 $\pm$ 1.51 | 49.80 $\pm$ 9.14 | 80.98 $\pm$ 3.65 | 90.15 $\pm$ 1.19 |
| | Width | 1.06 $\pm$ 0.40 | 6.54 $\pm$ 2.91 | 1.41 $\pm$ 0.31 | 3.71 $\pm$ 0.98 |
| Bouncing Ball dyn. | Cov | 92.54 $\pm$ 1.14 | 55.71 $\pm$ 8.03 | 81.66 $\pm$ 3.78 | 90.47 $\pm$ 2.30 |
| | Width | 1.18 $\pm$ 0.33 | 7.53 $\pm$ 3.16 | 1.47 $\pm$ 0.21 | 1.76 $\pm$ 0.71 |
| 3-Mode System | Cov | 89.21 $\pm$ 1.25 | 59.27 $\pm$ 3.74 | 80.54 $\pm$ 4.02 | 94.96 $\pm$ 1.96 |
| | Width | 0.06 $\pm$ 0.03 | 0.97 $\pm$ 0.27 | 0.25 $\pm$ 0.07 | 2.45 $\pm$ 0.72 |
| Traffic | Cov | 87.68 $\pm$ 1.28 | 68.83 $\pm$ 3.10 | 82.89 $\pm$ 2.43 | 92.38 $\pm$ 1.24 |
| | Width | 0.71 $\pm$ 1.92 | 12.53 $\pm$ 5.09 | 90.35 $\pm$ 21.44 | 7.92 $\pm$ 2.98 |
| Electricity | Cov | 95.63 $\pm$ 2.67 | 65.00 $\pm$ 9.99 | 82.00 $\pm$ 3.31 | 91.22 $\pm$ 1.29 |
| | Width | 193.25 $\pm$ 80.19 | 106.25 $\pm$ 159.82 | 83.42 $\pm$ 20.53 | 139.75 $\pm$ 620.44 |
| Dancing Bees | Cov | 84.52 $\pm$ 2.98 | 45.32 $\pm$ 8.49 | 70.52 $\pm$ 5.33 | 92.64 $\pm$ 3.19 |
| | Width | 0.03 $\pm$ 0.01 | 0.08 $\pm$ 0.01 | 0.22 $\pm$ 0.09 | 0.79 $\pm$ 0.27 |

**Longer horizon Data**   Result for the same set of experiments performed on longer horizon ($T = 10,000$ for synthetic datasets amd $T = 4,000$ for Electricity and Traffic datasets) is presented in table 5. We added the setting where the added noise to the bouncing ball dataset is varying over the course of the time series, which is more challenging than the original settings. In the longer horizon case, both SPCI and HopCPT achieve their state of the art performance. Our method remains valid yet achieves less sharp interval width.

CPTC offers distinct advantages over SPCI and HopCPT in data-limited regimes and applications requiring rapid deployment. While SPCI and HopCPT achieve excellent long-term performance on extensive time series (T $\geq$ 10,000) by learning residual patterns through quantile regression and Hopfield networks respectively, they require substantial data to train these models effectively during inference. Our results demonstrate that CPTC significantly outperforms these methods on shorter horizons (T $\leq$ 300), where insufficient data prevents SPCI and HopCPT from learning meaningful temporal correlations or residual patterns.

Table 5: Longer time horizon ($T = 10,000$ for synthetic datasets amd $T = 4,000$ for Electricity and Traffic datasets). Performance in synthetic and real-world datasets with target confidence $1 - \alpha = 0.9$. Methods that are *invalid* (coverage below 90%) are grayed out. Our method achieves a high level of calibration (coverage is close to 90%) consistently.

| | | RED-SDS | CP | ACI | SPCI | HopCPT | Ours |
|---|---|---|---|---|---|---|---|
| Bouncing Ball obs. | Cov | 18.53 $\pm$ 7.10 | 65.28 $\pm$ 5.89 | 89.25 $\pm$ 1.40 | 90.05 $\pm$ 0.04 | 90.02 $\pm$ 0.02 | 90.53 $\pm$ 0.23 |
| | Width | 1.65 $\pm$ 0.11 | 2.65 $\pm$ 0.85 | 3.53 $\pm$ 0.81 | 1.95 $\pm$ 0.12 | 1.99 $\pm$ 0.08 | 2.25 $\pm$ 0.54 |
| Bouncing Ball dyn. | Cov | 15.12 $\pm$ 15.50 | 42.88 $\pm$ 6.95 | 87.95 $\pm$ 1.05 | 90.04 $\pm$ 0.04 | 90.01 $\pm$ 0.04 | 90.05 $\pm$ 0.04 |
| | Width | 1.67 $\pm$ 0.13 | 0.95 $\pm$ 0.51 | 3.10 $\pm$ 1.04 | 3.24 $\pm$ 0.14 | 3.07 $\pm$ 0.10 | 3.18 $\pm$ 0.91 |
| 3-Mode System | Cov | 68.52 $\pm$ 3.53 | 65.24 $\pm$ 5.51 | 89.75 $\pm$ 0.90 | 90.05 $\pm$ 0.02 | 90.02 $\pm$ 0.04 | 90.08 $\pm$ 0.05 |
| | Width | 2.56 $\pm$ 0.21 | 2.33 $\pm$ 0.63 | 3.74 $\pm$ 0.61 | 2.67 $\pm$ 0.94 | 2.49 $\pm$ 1.31 | 3.12 $\pm$ 2.63 |
| Bouncing Ball obs. varying | Cov | 10.83 $\pm$ 22.53 | 40.25 $\pm$ 6.52 | 89.60 $\pm$ 1.25 | 90.02 $\pm$ 0.04 | 90.04 $\pm$ 0.00 | 90.10 $\pm$ 0.03 |
| | Width | 2.10 $\pm$ 0.45 | 1.73 $\pm$ 0.91 | 3.70 $\pm$ 0.95 | 2.11 $\pm$ 0.82 | 2.14 $\pm$ 0.73 | 2.41 $\pm$ 1.22 |
| Bouncing Ball dyn. varying | Cov | 14.25 $\pm$ 16.03 | 41.73 $\pm$ 7.12 | 88.05 $\pm$ 1.00 | 90.05 $\pm$ 0.03 | 90.02 $\pm$ 0.04 | 90.08 $\pm$ 0.03 |
| | Width | 1.85 $\pm$ 0.12 | 0.92 $\pm$ 0.55 | 3.13 $\pm$ 1.14 | 2.92 $\pm$ 0.21 | 3.05 $\pm$ 0.19 | 3.40 $\pm$ 0.32 |
| Traffic | Cov | 87.14 $\pm$ 10.51 | 70.52 $\pm$ 2.83 | 89.15 $\pm$ 0.38 | 90.01 $\pm$ 0.04 | 90.03 $\pm$ 0.03 | 90.02 $\pm$ 0.03 |
| | Width | 2.91 $\pm$ 1.62 | 0.43 $\pm$ 0.25 | 55.10 $\pm$ 24.13 | 4.81 $\pm$ 2.42 | 7.84 $\pm$ 10.53 | 8.55 $\pm$ 14.11 |
| Electricity | Cov | 76.10 $\pm$ 13.52 | 69.53 $\pm$ 3.21 | 89.05 $\pm$ 0.33 | 90.02 $\pm$ 0.03 | 90.00 $\pm$ 0.02 | 90.43 $\pm$ 0.03 |
| | Width | 91.30 $\pm$ 125.10 | 62.10 $\pm$ 451.20 | 31.40 $\pm$ 171.30 | 162.02 $\pm$ 11.40 | 161.70 $\pm$ 15.20 | 173.38 $\pm$ 31.10 |

**Aggregation methods.** In this study we compare the two different aggregation methods. First is the case where we chose the minimum size, i.e. $\Gamma_t(x_t) =$ Eqn 10. Algorithmically, we discretize the output space $\mathcal{Y}$ into intervals of length 0.02, and calculate probability mass for each interval. We refer to this method as *min*. Aggregation by union numbers (the same as in the main text using Eqn 11) are referred to as *union*. Results are present in table 6 where the performance are very similar. When there are two modes (bouncing ball cases) the prediction intervals are almost identical.

Table 6: Ablation Study on two different Union (*union*, eqn 11) and grid-discretization (*min*, eqn 10).

| | BB obs. | BB dyn. | 3-mode system | traffic | electricity | bees |
|---|---|---|---|---|---|---|
| Coverage (*union*) | 90.53 | 90.44 | 93.35 | 92.76 | 92.62 | 93.43 |
| Coverage (*min*) | 90.03 | 90.32 | 91.50 | 91.87 | 89.82 | 90.28 |
| Width (*union*) | 3.73 | 1.72 | 2.45 | 7.94 | 166.88 | 0.79 |
| Width (*min*) | 3.62 | 1.65 | 2.41 | 7.95 | 153.57 | 0.77 |

**Discussion on Misspecified Regimes.** We designed the number of latent states to be a hyperparameter given by the user. Our method ensures that the validity guarantee still holds even if the state cardinality is incorrect (theorem 4.2), as incorrect state cardinality can be formulated as a case of incorrect classification. Determining state space cardinality is a well-studied problem (see for example [14]), and we consider it to be beyond the scope of our paper.

Regarding the effect on performance, our ablation studies (Section 5.2, Tables 2 and 3) demonstrate that while the validity (coverage) of CPTC is robust to errors in state prediction, the sharpness (width of prediction intervals) can be affected. Table 3 explicitly shows that "the more accurate the state prediction is, the sharper the intervals." This implies that selecting a more appropriate number of regimes that better captures the true underlying dynamics would lead to narrower and more efficient prediction intervals.

## B.5   Qualitative Results

We provide additional visualizations to present qualitative comparisons of CPTC compared to baseline models. Figures 6, 7, and 8 show consistent results with the analysis in section 5, where CPTC shows stronger adaptivity and validity for uncertainty quantification in time series with these nonstationary dynamics.

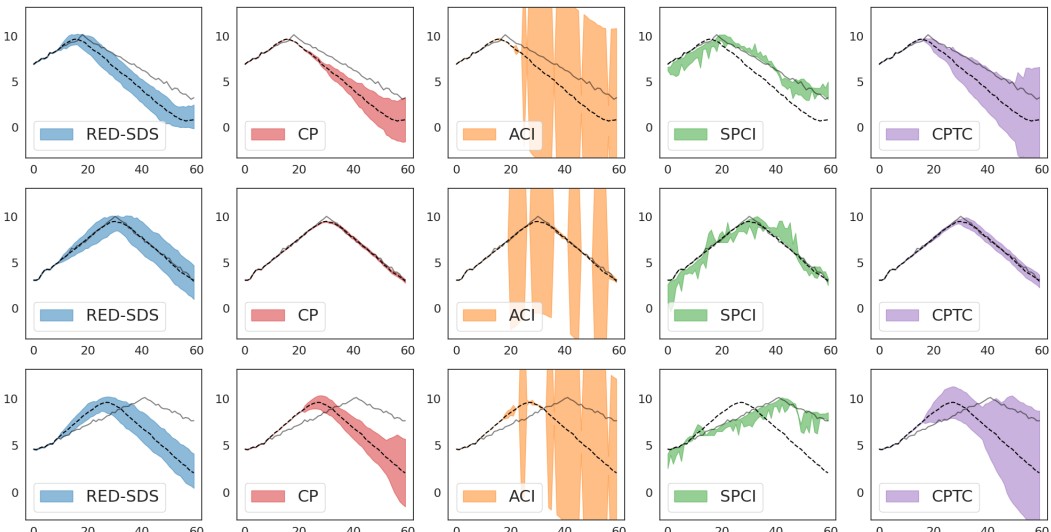

Figure 6: **Qualitative comparison of the prediction interval on the bouncing ball w/ dynamics noise dataset.** The dashed line is the predicted state whereas the solid gray is the observation. We see that our CPTC algorithm can increase the interval in anticipation of an uncertain state change (top panel), is stabler when the prediction is accuracy compared to ACI and SPCI (middle panel), and is adaptive to inaccurate predictions (botom panel).

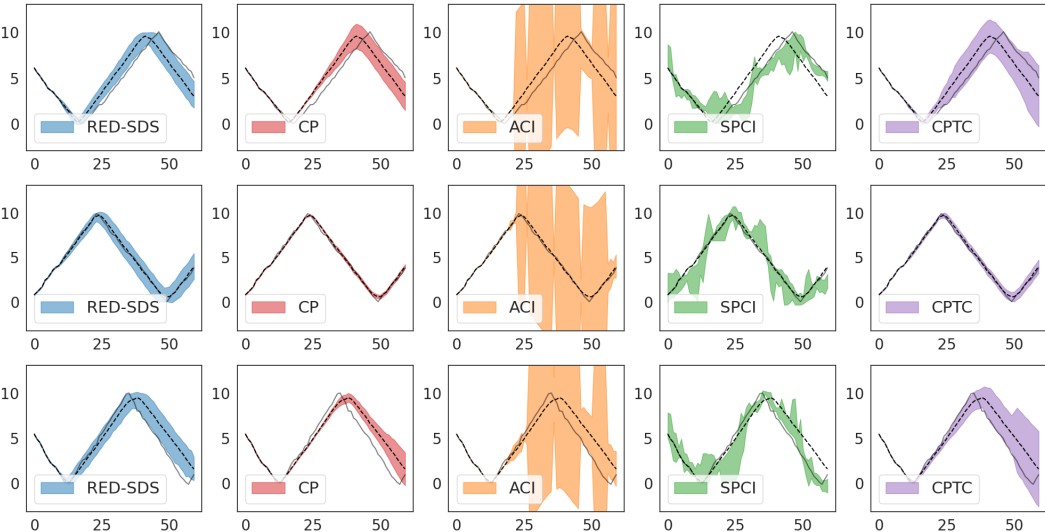

Figure 7: Visualization of prediction intervals on the Bouncing Ball Obs dataset.

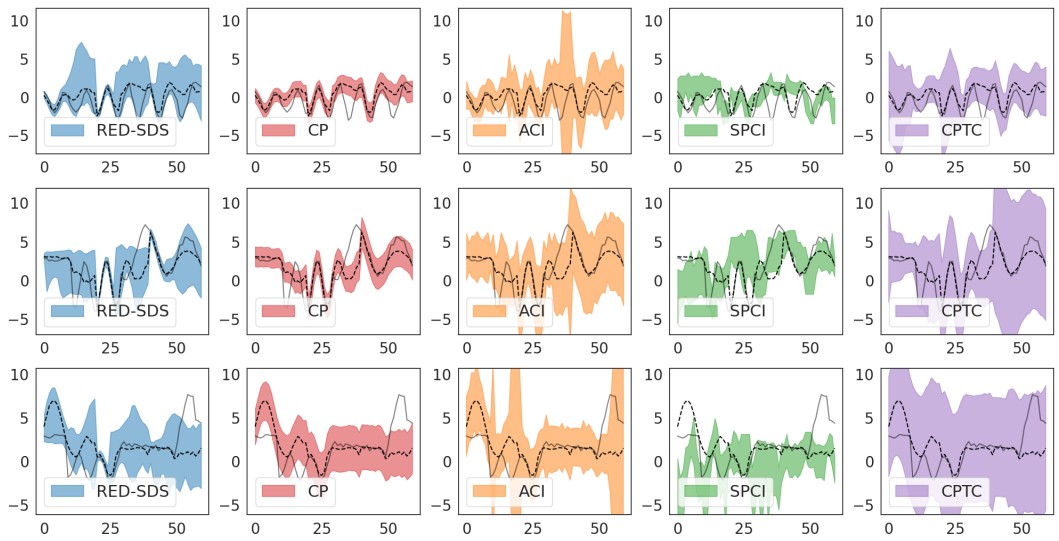

Figure 8: Visualization of prediction intervals on the 3 mode system datasets.

## B.6 Runtime analysis

We provide a runtime analysis comparing the aggregation methods in Equations 10 and 11 and with baselines.

**Notation**

- $T$: Total timesteps, $K$: Number of states, $M$: State prediction model inference complexity
- Score sets $\mathcal{S}_z$ maintained in sorted order (quantile computation in $\mathcal{O}(1)$, insertion $\mathcal{O}(\log T)$)

**Complexity Per Timestep**

1. Computing quantile $Q^{1-\alpha_t}(\mathcal{S}_z^{(l)})$ for each state, totaling $\mathcal{O}(K)$.
2. State Prediction: computing $\hat{p}(z_t|x_t)$ and sampling a state, totaling $\mathcal{O}(M)$.
3. Aggregation
    - *Eq. 10, Weighted Level-Set:* $\mathcal{O}(K \cdot \lceil 1/\delta \rceil^d)$ where $\delta$ is discretization resolution
    - *Eq. 11, Union:* $\mathcal{O}(K \log K)$ for sorting states by probability
4. Updates: $\mathcal{O}(\log t)$ for inserting new score into sorted set.

*Therefore, the total complexity of CPTC with horizon $T$ is:*

$\mathcal{O}(T \cdot (M + K \cdot \lceil \frac{1}{\delta} \rceil^d) + T \log T)$ if we use the Weighted Level-Set as aggregation in Eq. 10, and

$\mathcal{O}(T \cdot (M + K \log K) + T \log T)$ if we use the more efficient Union aggregation in Eq. 11.

In practice, both $M$ (inference cost) and $K$ (number of discrete states) are small. For applications where $K = \mathcal{O}(1)$ and $M = \mathcal{O}(1)$, CPTC achieves near-linear scaling $\mathcal{O}(T \log T)$, the same as ACI (see table below).

If we use the more precise weighted level set aggregation method, the runtime largely depends on the discretization resolution $\delta$, output dimension $d$, and the number of states $K$. We remark the level set approach scales poorly if the resolution, dimension, or number of states are high.

| Method | Time Complexity | Remarks |
|---|---|---|
| ACI | $\mathcal{O}(T \log T)$ | Single score set, no state awareness |
| SPCI / HopCPT | $\mathcal{O}(T^2 \log T)$ | Re-train regression model at every time step |
| CPTC (Union) | $\mathcal{O}(T \log T)$ | State-aware, pre-trained predictor |

