# OpenReview forum: "Conformal Prediction for Time-series Forecasting with Change Points"
_NeurIPS.cc/2025/Conference — NeurIPS 2025 poster_

### Official Review · Reviewer_3vb3 · 2025-06-26

**Clarity:** 4
**Significance:** 3
**Originality:** 3
**Rating:** 4
**Confidence:** 4

**Summary:**

The author introduced CPTC method, which is a conformal prediction method for time series that is based on switching dynamical systems. Supposing that the time series can be classified into different states, the CPTC method split the dataset into subsets and combine the prediction set with the state predictor. The method is able to quickly respond to the change points in the time series and generate valid confidence interval for each state. In theory, the authors prove the asymptotic validity under the assumption that the true state distribution follows an underlying stationary distribution. In experiment, the authors compare the method with several common bayesian or CP methods to show that their method is able to achieve valid coverage in the change point setting while the other methods cannot.

**Questions:**

1. Could authors explain when the assumption for the time series holds?

2. Could authors explain how to select the number of possible regimes in predicting the states? Would that affect the performance of the method?

3. If the state predictor works very poorly, will the performance of the method be similar to the performance of ACI?

**Ethical Concerns:**

["NO or VERY MINOR ethics concerns only"]

**Limitations:**

yes

**Quality:**

3

**Strengths And Weaknesses:**

Strength:
1. Writing is clear
2. Have solid experiments, compare with several common benchmark methods on 3 synthetic datasets and 3 real datasets
3. Show the method is asymptotic valid under certain assumption in theory, also show the method is robust and can quickly react to change point
4. Have novelty in method


Weakness:
1. The method assumes that the time series is switching between some true states, and requires a state predictor to build the prediction interval.
Need more discussion about when the assumption holds and what to do when the assumption fails to hold.

---

> ### Author Response · Authors · 2025-07-31
> **Rebuttal**
>
> We thank the reviewer for their positive assessment of our work, acknowledging the clarity, experimental rigor, theoretical soundness, and novelty of our method. We address the points raised below.
>
> > Weakness1 / Q1. Could authors explain when the assumption for the time series holds?
>
> The SDS assumption (Equation 5) is appropriate for time series that exhibit regime-switching behavior - i.e., periods where the underlying data-generating process remains relatively stable, punctuated by abrupt shifts to different regimes. This includes:
>
> - Naturally regime-switching domains: Electricity demand (day/night cycles), traffic flow (weekday/weekend patterns), financial markets (volatility regimes), and manufacturing systems (operational modes)
> - Time series with predictable change points: Where shifts correlate with observable covariates or temporal patterns
> - Systems with discrete operational states: Industrial processes, biological systems with distinct phases, or any domain where the underlying process can be characterized by a finite set of behaviors
>
> Importantly, our method doesn't require perfect adherence to the SDS assumption. Theorems 4.2-4.3 show robustness to (1) imperfect state predictions (Theorem 4.3) (2) model misspecification, and (3) unknown shift mechanisms. This means that even if the learned states are not perfectly aligned with true underlying "modes," or if the assumption of distinct switching states is only an approximation, our adaptive calibration mechanism ensures overall coverage validity, as experiments show in Section 5.2.
> The trade-off, as shown in Table 3 (page 9), is that less accurate state predictions might lead to wider prediction intervals, even if coverage remains valid. This highlights the practical utility of our method: it offers a robust solution even when the "true" underlying states are not perfectly known or precisely modeled.
>
> We will highlight the above points in the paper’s discussion section to further clarify. We thank the reviewer for raising this important discussion.
>
> > 2. Could authors explain how to select the number of possible regimes in predicting the states? Would that affect the performance of the method?
>
> We designed the number of latent states to be a hyperparameter given by the user. Our method ensures that the validity guarantee still holds even if the state cardinality is incorrect (theorem 4.2), as  incorrect state cardinality can be formulated as a case of incorrect classification. Determining state space cardinality is a well-studied problem (see for example [3]), and we consider it to be beyond the scope of our paper.
>
> Regarding the effect on performance, our ablation studies (Section 5.2, page 9, Tables 2 and 3) demonstrate that while the validity (coverage) of CPTC is robust to errors in state prediction, the sharpness (width of prediction intervals) can be affected. Table 3 explicitly shows that "the more accurate the state prediction is, the sharper the intervals." This implies that selecting a more appropriate number of regimes that better captures the true underlying dynamics would lead to narrower and more efficient prediction intervals.
>
> [3] Branicky, Michael S. "Stability of switched and hybrid systems." Proceedings of 1994 33rd IEEE conference on decision and control. Vol. 4. IEEE, 1994.
>
> > 3. If the state predictor works very poorly, will the performance of the method be similar to the performance of ACI?
>
> Yes, exactly. There will be Z (depends on the number of states specified by the user) concurrent ACI processes.
>
> ---
> We thank the reviewer again for the thoughtful review; please let us know if our response answers your questions.

---

> ### Comment · Reviewer_3vb3 · 2025-08-02
>
> Thank the authors for their careful answers, and I will keep my rating of 4.

---

### Official Review · Reviewer_hm7E · 2025-06-26

**Clarity:** 3
**Significance:** 2
**Originality:** 2
**Rating:** 4
**Confidence:** 5

**Summary:**

This paper presents a novel method (CPTC) that integrates switching dynamical systems (SDS) with online conformal prediction to handle uncertainty quantification in non-stationary time series with change points. The approach is theoretically grounded and empirically validated, addressing a significant gap in existing CP methods.

**Questions:**

1. In table 1, why is the coverage of ACI invalid? It seems that both theoretical and empirical results in other papers (see \[1] and \[2]) show that ACI will achieve long-term coverage despite fluctuations and inefficient intervals. Does this correlates to the choice of learning rates $\gamma$? The authors should dicuss it in depth.

2. In the experimental section, most of the baseline models used for comparison are relatively outdated. It is recommended that the authors compare their model with more up-to-date online CP methods tackling changepoints, such as \[2] and \[3].

3. CPTC assumes the system consists of several underlying states, such as day/night. However, in the area of conformal prediction, it seems more appropriate to use group-conditional-based methods like \[4] and the baseline should add such methods. How do the authors comment about this?

[1] Gibbs I, Candes E. Adaptive conformal inference under distribution shift, NIPS 2021;

[2] Wu J, Hu D, Bao Y, et al. Error-quantified conformal inference for time series, ICLR 2025;

[3] Angelopoulos A N, Barber R, Bates S. Online conformal prediction with decaying step sizes, ICML 2024;

[4] Zecchin M, Simeone O. Localized Adaptive Risk Control, NIPS 2024.

**Ethical Concerns:**

["NO or VERY MINOR ethics concerns only"]

**Final Justification:**

Most of my concerns have been addressed. Since this work is relatively derivative, so I give rate 4 (Borderline accept).

**Limitations:**

yes

**Quality:**

2

**Strengths And Weaknesses:**

**Strengths:**

 1. A new method of conformal prediction for time series with changepoints, leveraging an SDS model.
 2. The mathematical proofs are easy to follow and the notations are clear.
 3. Many online CP methods similar to ACI can be incorporated into the framework.
 4. The authors demonstrate the practical utility of their methods with an extensive series of experiments.

**Weaknesses:**

 1. The CPTC algorithm proposed in this paper simply integrates an SDS model into the online CP framework. The switching scenarios are also relatively limited, and the authors could consider the switching mechanism's effect on online CP in greater depth.
 2. Incomplete experiments. See questions.

---

> ### Author Response · Authors · 2025-07-31
> **Rebuttal**
>
> We thank the reviewer for their insightful comments and appreciation of our work. We address the points raised below:
>
> > 1. In table 1, why is the coverage of ACI invalid? It seems that both theoretical and empirical results in other papers (see [1] and [2]) show that ACI will achieve long-term coverage despite fluctuations and inefficient intervals. Does this correlate to the choice of learning rates ? The authors should discuss it in depth.
>
> We appreciate this crucial question. The authors of ACI themselves acknowledge failures in their original ACI method in their follow up work [5]: "Previous approaches to this problem [including their original ACI] suffer from over-weighting historical data and thus may fail to quickly react to the underlying dynamics." ACI’s slowness to adapt to abrupt changes is also analyzed in [6]. ACI needs a long horizon for it to balance the under-coverage with over-coverage, as seen in the case of [2] where T= 2000, and in [1] where T=3000 (election) or 5000+ (stocks). As expected, in the case of longer time horizons, ACI achieves near-valid coverage on our datasets (table 4 in Appendix).
>
> We mention in our analysis (lines 305-306, page 7), "In the pretense of change points, ACI's coverage fluctuates dramatically and does not converge within the horizon." While ACI is asymptotically valid, the coverage can fluctuate significantly in shorter time horizons, especially in the presence of abrupt distribution shifts or change points, as observed in our experiments.
>
> This brings us to the main contribution of CPTC: We show that fast adaptation is important for shorter time series (T = 200 for our synthetic datasets, 400 for electricity and traffic, and 60 for bees), where the ability to *anticipate distribution shifts* can significantly improve both coverage and efficiency. Time series of < 500 length is common in practice (health time series, user activity of web services or apps, and physical/chemical/biological systems with distinct phases), and we developed CPTC to be a practical and sound CP variant for these situations.
>
> [5] Angelopoulos, Anastasios, Emmanuel Candes, and Ryan J. Tibshirani. "Conformal pid control for time series prediction." Advances in neural information processing systems 36 (2023): 23047-23074.
>
> [6] Gibbs, Isaac, and Emmanuel J. Candès. "Conformal inference for online prediction with arbitrary distribution shifts." Journal of Machine Learning Research 25.162 (2024): 1-36.
>
> > 2. The experiment section is outdated.
>
> We thank the reviewer for bringing these two important works to our attention. We have now added these baselines to our experimental comparison in Table 1 (see table below). We will add more analysis with visualizations in the finalized version of our paper.
>
> We would like to emphasize that these methods, just like the ones reported in table 1 of the paper, are *reactive*. The question we try to address with CPTC is - what happens when we can anticipate distribution shift? CPTC is designed such that it greatly improves coverage compared to purely reactive methods, while still maintaining the ability to adapt to unknown shifts. To our knowledge, there are no other models that explore the same problem; CPTC provides practitioners with a simple and sound solution, when they have the ability (such as via a state space model) to predict shifts.
>
> > 3. CPTC assumes the system consists of several underlying states, such as day/night. However, in the area of conformal prediction, it seems more appropriate to use group-conditional-based methods like [4] and the baseline should add such methods. How do the authors comment about this?
>
> Thank you for bringing this work to our attention! We have added L-ARC (with miscoverage as loss function) as a baseline in the table below. We see that in the case of shorter time series with switching uncertainty dynamics CPTC outperforms the baseline.
>
>  L-ARC shares commonalities to our setting and result, but it differs in that:
>
> The localization of L-ARC comes from adjusting the threshold using a kernel function for the inputs $X_t$ (RBF kernel in most of the paper’s experiments). That means (1) the kernel weighting would need to be learned online, and the more complex/localized the landscape is, the more samples the algorithm needs, and (2) the kernel needs to be hyper parameter-tuned to fit the data’s latent space. These properties make L-ARC less sample-efficient compared to our method, and more similar to the online regression-based CP methods like HopCPT and SPCI. These works provide approximate (sample-)conditional coverage guarantees similar to L-ARC; CPTC merely provides group-conditional guarantees.
>
> ---
>
> We thank the reviewer again for the thoughtful review; please let us know if our response answers your questions.

---

> > ### Author Response · Authors · 2025-07-31
> > **Table**
> >
> > |                       |        | DtACI [3]       | ECI-cutoff [4]  | L-ARC [5]         | Ours              |
> > |-----------------------|--------|-----------------|------------------|-------------------|-------------------|
> > | Bouncing ball obs     | cov    | 82.97 ± 1.35     | 79.86 ± 1.42      | 75.12 ± 9.25      | **91.03 ± 1.37**  |
> > |                       | width  | 3.21 ± 0.80      | 3.02 ± 0.72       | 1.89 ± 0.76       | 2.13 ± 2.15       |
> > | Bouncing ball Dyn     | cov    | 79.63 ± 1.10     | 80.91 ± 1.28      | 78.63 ± 6.21      | **90.44 ± 1.51**  |
> > |                       | width  | 3.51 ± 2.20      | 2.32 ± 0.19       | 2.44 ± 0.23       | 4.92 ± 2.79       |
> > | 3-mode system         | cov    | 88.82 ± 0.95     | 83.67 ± 0.89      | **92.58 ± 2.77**  | **93.35 ± 3.51**  |
> > |                       | width  | 12.55 ± 7.52     | 2.39 ± 0.48       | 12.31 ± 2.85      | 13.68 ± 2.49      |
> > | Traffic               | cov    | 89.14 ± 0.40     | 86.12 ± 0.50      | **90.05 ± 6.24**  | **94.76 ± 4.93**  |
> > |                       | width  | 10.35 ± 9.85     | 7.02 ± 9.72       | 8.39 ± 10.13      | 8.34 ± 14.44      |
> > | Electricity           | cov    | 88.35 ± 0.38     | 89.41 ± 0.47      | 86.42 ± 4.96      | **92.62 ± 6.14**  |
> > |                       | width  | 189.51 ± 187.25  | 138.72 ± 83.30    | 157.82 ± 95.02    | 166.88 ± 129.07   |
> > | Dancing Bees          | cov    | 78.91 ± 3.02     | 80.12 ± 2.85      | 75.92 ± 14.21     | **93.43 ± 11.42** |
> > |                       | width  | 0.84 ± 0.21      | 0.78 ± 0.22       | 0.98 ± 0.25       | 1.15 ± 0.56       |

---

> > ### Comment · Reviewer_hm7E · 2025-08-05
> >
> > Thanks for the detailed feedback. Most of my concerns have been addressed and I will increase my score to 4. Good luck!

---

### Official Review · Reviewer_Nmg7 · 2025-06-26

**Clarity:** 2
**Significance:** 2
**Originality:** 3
**Rating:** 4
**Confidence:** 4

**Summary:**

The manuscript describes an algorithm for a conformal prediction to quantify uncertainty for time series data, whose data generating distribution changes over time. The proposed method relies on a pre-trained switching state model to identify the hidden state at each time step and build a conformal prediction model as a mixture distribution.

**Questions:**

1. The probabilistic model shown in (5) is inconsistent with the rest of the paper.
  - the model in (5) is a generative model that leans the distribution. However, rest of the paper, the baseline model looks like a deterministic forecast model.
  - in most of the places, the conformal prediction is computed based on $p(z_t|x_t)$, which is inconsistent with the probabilistic model in (5). Moreover, $p(z_t|x_t)$ indicates the latent state is determined by the data in the current time step, $x_t$. However, in the time series problems, in general, given only one time step value, $x_t$, the system state cannot be uniquely determined. Did you mean $p(z_t|x_0,\cdots,x_t)$?

2. There are two aggregation schemes Eqns (10) and (11). And the authors ague that both methods results in similar prediction interval based on observations on a few data sets. However, in all of their experiments, the number of latent state is very small, 2 or 3. So, it is unclear if the behavior can be extrapolated to a larger number of latent states.

3. Similar to Q2, what's the effects of the size of the latent state? How do you know what's going to be the right number of latent states for each data set?

4. What is $E[err_t]$ in Theorem 4.2? Expectation over what distribution?

5. In Appendix B, it is mentioned that the forecast was made with a "deterministic latent states" ($z_t$) and "stochastic systems states" ($x_t$). This seems not consistent with the probabilistic model assumed in the paper. What is "deterministic latent states"? Can you prove that it gives a correct Monte Carlo estimate of the given probabilistic model?

6. What's the effect of the data set size?

7. It looks like the proposed method works for univariate time series. Can it be used for a multi-variate time series? Then, how to consider the covariance?

8. The organization can be improved. In particular, online CP is introduced in section 3.1. But here the authors only describe the theoretical guarantee without outlining the core method. And in section 4.1 the online CP is described together with the new features. This organization is a little bit confusing. I would suggest the authors to move the description of the online CP to section 3.1 and only focus on novel features in section 4.1.

**Ethical Concerns:**

["NO or VERY MINOR ethics concerns only"]

**Final Justification:**

I believe that the manuscript is interesting enough and the proposed method can be useful for time series prediction models, many of which are deterministic. The authors addressed most of my concerns. I am still not confident about the experiments and hyperparameter studies, but the study is worth reporting to the community.

**Limitations:**

See Questions.

**Quality:**

3

**Strengths And Weaknesses:**

Strength:
The paper is straightforward and easy to read the core idea and the proposal to tackle the challenges. The proposed method is model-agnostic as long as the baseline model provides a discrete latent space.

Weakness:
The experiments are weak, and there are many places that need clarification / further explanation. See the questions below

---

> ### Author Response · Authors · 2025-08-01
> **Rebuttal**
>
> We thank the reviewer for their insightful comments and appreciation of our work. We address the points raised below:
>
> ---
> > 1. Probabilistic model inconsistency
>
> We choose the mean of the forecaster output as the deterministic forecast. This is common practice in time series CP works, such as the use of DeepAR in [1,2]. We will clarify the usage in the paper per the reviewer’s suggestion.
> Yes, thank you for your correction. We will use P(z_t|x_{0:t}) to indicate the predicted probability of states in our paper, in order to stay consistent with the SDS model introduced in the background section.
>
> [1] Xu, Chen, and Yao Xie. "Conformal prediction interval for dynamic time-series." International Conference on Machine Learning. PMLR, 2021.
> [2] Xu, Chen, and Yao Xie. "Sequential predictive conformal inference for time series." International Conference on Machine Learning. PMLR, 2023.
>
> > 2. Larger number of latent states vs. two aggregation metrics.
>
> This is a great observation. We found that the difference between the two aggregation methods does not depend on the number of states, but the discrepancy of (1) behavior and (2) magnitude of uncertainty between the most likely states. In the case where there are many states and the behaviors vary a lot, equation (11) outputs over-covering large intervals during state transitions. As this result is hard to present with tables, we will include qualitative visualizations in the finalized version.
>
> > 3. What are the effects of the size of the latent state? How do you know what's going to be the right number of latent states for each data set?
>
>
> We designed the number of latent states to be a hyperparameter given by the user. Our method ensures that the validity guarantee still holds even if the state cardinality is incorrect (theorem 4.2), as  incorrect state cardinality can be formulated as a case of incorrect classification. Determining state space cardinality is a well-studied problem (see for example [3]), and we consider it to be beyond the scope of our paper.
>
> Please see table below for results on our Z-mode system synthetic setting with misspecified states - when Z is misspecified, the uncertainty sets becomes larger, in line with the behavior we see for state predictors with large misclassification rate in Appendix B.4.
>
> [3] Branicky, Michael S. "Stability of switched and hybrid systems." Proceedings of 1994 33rd IEEE conference on decision and control. Vol. 4. IEEE, 1994.
>
> > 4. The expectation notation taken in Theorem 4.2
>
> The expectation is taken over the data generation process at time $t$ for $(x_t, y_t)$. The notation is consistent with derivation in the ACI paper by Gibbs & Candes. it simply means $E[err_t] =P (Y_t\not\in \Gamma_t (\alpha_t))$. We will clarify before introducing the expectation in our proofs.
>
> > 5. “Deterministic latent states” and “stochastic continuous states”
>
> We thank the reviewer for capturing this error, it was a typo during writing. We will change the sentence at line 600-602 to “We sample $N=100$ of ${(z_t, x_t, y_t)}_{t=1}^k$ triplet trajectories from the trained model by Monte Carlo sampling, and use the samples to calculate prediction quantiles for table 1, in accordance to the original paper.”
>
> > 6. What is the effect of the data set size?
>
> Our model is purely online, so the effective data set is the warm start window $w$ and the total time series length. We study the effect of a longer horizon and warm start in section B.4.
>
> Intuitively, larger training data will result in a better prediction and state-transition model. We have proven in the paper that the validity guarantee will hold regardless of the performance of the base models.
>
> > 7. Multivariate time series
>
> Our method does apply to multivariate time series. The Bee dataset is an example - the time series is 6 dimensional, and the prediction is on the first 2 dimensions. Under the conformal prediction framework, the nonconformity score is always a scalar.  Many papers explore the topic of tailoring CP to be efficient for multivariate data [4,5]. They are orthogonal to our method and can be used in combination. To incorporate the reviewer's suggestion, we will add the above paragraph to the appendix to clarify.
>
> [4] Xu, Chen, Hanyang Jiang, and Yao Xie. "Conformal prediction for multi-dimensional time series by ellipsoidal sets." arXiv preprint arXiv:2403.03850 (2024).
>
> [5] Messoudi, Soundouss, Sébastien Destercke, and Sylvain Rousseau. "Copula-based conformal prediction for multi-target regression." Pattern Recognition 120 (2021): 108101.
>
> > 8. Organization
>
> We appreciate the reviewer’s suggestion. Equation 9 was included in section 4.1 for clarity and completeness, but we will introduce the online update algorithm in the background section as well.
>
> ----
> We thank the reviewer again for the thoughtful review; please let us know if our response answers your questions.

---

> > ### Author Response · Authors · 2025-08-01
> > **Table**
> >
> > | Input Z / True Z || Z = 3| Z = 6 |
> > |----------|---|---------|----------|
> > |   \hat{Z} = 3       |   cov|  93.35 ± 3.51     |   92.89 ± 1.51       |
> > |          |  width         | 13.68 ± 2.49  |     15.01 ± 3.63     |
> > |  \hat{Z} = 6        | cov |  89.23 ± 2.01|   91.26 ± 1.77       |
> > |          |  width          |  17.25 ± 3.44   |   15.42 ± 2.06       |

---

> > ### Comment · Reviewer_Nmg7 · 2025-08-05
> >
> > Regarding Item 7: While the method can be applied to a multivariate time series, it looks like CP is based on the marginal distribution of each channel, not the full joint distribution. It should be clarified.
> >
> > My concerns were more about the clarity of the manuscript. Assuming the authors will make the changes as outlined in the rebuttal, I am updating my score.

---

> > > ### Author Response · Authors · 2025-08-07
> > >
> > > Yes - that is a fair point. Wether the coverage guarantee is over the joint distribution or individual channels depends on the selection of  the nonconformity scores, with their own tradeoffs. We will clarify this point in the paper, thank you for your suggestion.
> > >
> > > Again, we really appreciate the reviewer's constructive feedback and engagement, which has helped enhance the overall quality of this work.

---

### Official Review · Reviewer_pacc · 2025-07-02

**Clarity:** 3
**Significance:** 2
**Originality:** 3
**Rating:** 4
**Confidence:** 4

**Summary:**

This paper proposes a new approach to conformal prediction for time series data with change points. The proposed method leverages switching dynamical systems to model different underlying dynamics in online environments. For each state, a separate prediction interval is constructed, and a final prediction set is formed by aggregating these state-specific intervals based on the predicted state distribution.

**Questions:**

1-	Could the authors provide the computational complexity of the proposed method and compare it with that of previous approaches?

2-	Could the authors clarify how the warm-start data is selected? Additionally, is there any empirical evaluation of its impact on coverage and prediction set size?

3-	Could the authors clarify how the state transition model is constructed? Specifically, is the transition matrix learned offline, and is the probability of switching between states kept fixed during all test-time steps?

4-	The proposed method creates state-specific prediction intervals for each latent state and then aggregates them. I would kindly ask the authors to compare their approach with prior methods that also generate multiple prediction sets and then select or aggregate among them. For example, ACI’s sensitivity to learning rate has been addressed in [1], and other methods such as [2] construct prediction sets for each learning model and then select or combine them.

[1] Gibbs, I. and Candès, E.J., 2024. Conformal inference for online prediction with arbitrary distribution shifts. Journal of Machine Learning Research, 25(162), pp.1-36.

[2] Hajihashemi, E. and Shen, Y., 2024. Multi-model ensemble conformal prediction in dynamic environments. arXiv preprint arXiv:2411.03678.

**Ethical Concerns:**

["NO or VERY MINOR ethics concerns only"]

**Final Justification:**

The idea proposed in this paper is novel. During the rebuttal, the authors addressed most of my concerns, such as providing the computational complexity and comparing the proposed method with previous approaches; therefore, I have decided to increase my confidence in the score. I suggest that the authors include the computational complexity in the final version of the manuscript.

**Limitations:**

Limitations are mentioned in the conclusion

**Quality:**

2

**Strengths And Weaknesses:**

Strengths:  The paper presents a novel idea with solid theoretical guarantees. The motivation is clearly stated, and the paper is well-written and easy to follow.

Weakness: Please check the Questions.

---

> ### Author Response · Authors · 2025-07-31
> **Rebuttal**
>
> > 1. Could the authors provide the computational complexity of the proposed method and compare it with that of previous approaches?
>
> We provided computation complexity comparisons in Appendix B.2, as follows:
>
> “All experiments are done on a server machine with an Nvidia A100 GPU, with some data processing 607 and analytics performed on an Apple Macbook Pro laptop computer with M1 chip. As running time varies greatly depending on implementation and hardware, we provide a rough range of algorithm 609 runtime. Training the RED-SDS base model (1 seed) takes 2-5 hours for our datasets, and inference takes around 15-20 minutes for all datasets. Among the conformal prediction baselines, SPCI is the most computationally demanding, taking 8-10 hours for inference on our benchmarks. HopCPT follows, taking around 1 hour. CP, ACI, and our method CPTC do not require extensive inference-time computation, completing all inference within 5 minutes”.
>
> Being computationally light and fast is a strength of CPTC - no fitting or re-fitting of residual prediction models is needed after each time step, as in the case of previous works HopCPT and SPCI.
>
> > 2. How is warm start data selected?
>
> The warm start window is the first $w$ steps of each test time series. There is no selection in the sense of choosing some data over others. We report the metrics averaged over horizon $T$ and then over the N test samples.
>
> he length of the warm-start window, $w$, is 25% of the length of the time series in our datasets. In baseline methods, the warm start window is typically very large - HopCPT uses 15-33% of time series of length in the thousands to tens of thousands for calibration.
>
> > 3. Could the authors clarify how the state transition model is constructed? Specifically, is the transition matrix learned offline, and is the probability of switching between states kept fixed during all test-time steps?
>
> Yes! The discrete latent states are extracted from a base model that is learned offline. Namely, we use the RED-SDS model ([8] in the paper) for all of our data sets because of its performance on time series datasets. The state transitions are not updated online, and the adaptive updates to the conformal thresholds are designed primarily to correct for state prediction errors. We do not assume the probability of switching states is kept the same, but we do assume the resulting state distribution to be stationary (assumption 1 in paper).
>
> The motivation of our algorithm is that, for many systems that require UQ, the model already models a discrete state. In this paper we study how such information can be utilized to anticipate distribution shifts in prediction behavior and thereby improve validity of UQ. Studying the more complex setting of updating/adapting switching states would be a great direction for future work.
>
> > 4. Compare to previous works [1] and [2]
>
> We thank the reviewer for bringing these two works into our attention. We have added [1] as a baseline (see table below), and will provide more analysis with visualizations in the finalized version of our paper.
>
> [2] (Hajihashemi and Shen, 2024) introduces a multi-model ensemble conformal prediction (SAMOCP) framework,  and the focus of their work is on optimizing model selection within a pool of experts, combining them to achieve coverage. This is different from our setting where the SDS is a single integrated model. We acknowledge the similarity in the motivation and will cite [2] in our literature review.
>
> The drawback of both these methods compared to CPTC is that they are *reactive*. The question we try to address with CPTC is - what happens when we can *anticipate* distribution shift? CPTC is designed such that it greatly improves coverage compared to purely reactive methods, while still maintaining the ability to adapt to unknown shifts. To our knowledge, there are no other models that explore the same problem; CPTC provides practitioners with a simple and sound solution, when they have the ability (such as via a state space model) to predict shifts.

---

> > ### Author Response · Authors · 2025-07-31
> > **table**
> >
> > |                       |        | DtACI [1]       | Ours              |
> > |-----------------------|--------|-----------------|-------------------|
> > | Bouncing ball obs     | cov    | 82.97 ± 1.35     | **91.03 ± 1.37**  |
> > |                       | width  | 3.21 ± 0.80      | 2.13 ± 2.15       |
> > | Bouncing ball Dyn     | cov    | 79.63 ± 1.10     | **90.44 ± 1.51**  |
> > |                       | width  | 3.51 ± 2.20      | 4.92 ± 2.79       |
> > | 3-mode system         | cov    | 88.82 ± 0.95     | **93.35 ± 3.51**  |
> > |                       | width  | 12.55 ± 7.52     | 13.68 ± 2.49      |
> > | Traffic               | cov    | 89.14 ± 0.40     | **94.76 ± 4.93**  |
> > |                       | width  | 10.35 ± 9.85     | 8.34 ± 14.44      |
> > | Electricity           | cov    | 88.35 ± 0.38     | **92.62 ± 6.14**  |
> > |                       | width  | 189.51 ± 187.25  | 166.88 ± 129.07   |
> > | Dancing Bees          | cov    | 78.91 ± 3.02     | **93.43 ± 11.42** |
> > |                       | width  | 0.84 ± 0.21      | 1.15 ± 0.56       |

---

> > ### Comment · Reviewer_pacc · 2025-08-05
> >
> > Thank you to the authors for the detailed responses. However, I believe Question 1 has not been addressed yet. I was asking for a theoretical analysis of the computational complexity.

---

> ### Author Response · Authors · 2025-08-06
>
> I see! sorry for the misunderstanding. We provide a comprehensive analysis below, comparing the aggregation methods in Equations 10 and 11 and with baselines.
>
> *Notation*
> - $T$: Total timesteps, $K$: Number of states, $M$: State prediction model inference complexity
> - Score sets $\mathcal{S}_z$ maintained in sorted order (quantile computation in $\mathcal{O}(1)$, insertion  $\mathcal{O}(\log T)$)
>
> *Complexity Per Timestep*
>
> 1. Computing quantile $Q^{1-\alpha_{z,t}}(\mathcal{S}_z^{(t)})$ for each state, totaling $\mathcal{O}(K)$.
> 2. State Prediction: computing $\hat{p}(z_t|x_t)$ and sampling a state, totaling $\mathcal{O}(M)$.
> 3. Aggregation
> - *Eq. 10, Weighted Level-Set:* $\mathcal{O}(K \cdot \lceil 1/\delta \rceil^d )$ where $\delta$ is discretization resolution
> - *Eq. 11, Union:* $\mathcal{O}(K \log K)$ for sorting states by probability
> 4. Updates: $\mathcal{O}(\log t)$ for inserting new score into sorted set.
>
>
> *Therefore, the total complexity of CPTC with horizon $T$ is:*
>
>
> $\mathcal{O}(T \cdot (M + K \cdot \left\lceil \frac{1}{\delta} \right\rceil^d)+ T \log T)$ if we use the Weighted Level-Set as aggregation in Eq. 10, and
>
> $\mathcal{O}(T \cdot (M  + K \log K) + T \log T)$ if we use the more efficient Union aggregation in Eq. 11.
>
> ---
>
> In practice,  both $M$ (inference cost) and $K$ (number of discrete states) are small. For applications where $K = \mathcal{O}(1)$  and $M = \mathcal{O}(1)$, CPTC achieves near-linear scaling $\mathcal{O}(T \log T)$, the same as ACI (see table below).
>
> If we use the more precise weighted level set aggregation method, the runtime largely depends on the discretization resolution  $\delta$, output dimension $d$, and the number of states $K$. We remark the level set approach scales poorly if the resolution, dimension, or number of states are high.
>
> | Method | Time Complexity | Remarks |
> |--------|----------------|-------------------|
> | ACI | $\mathcal{O}(T \log T)$ | Single score set, no state awareness |
> | SPCI / HopCPT | $\mathcal{O}(T^2 \log T )$ | Re-train regression model at every time step  |
> | CPTC (Union) | $\mathcal{O}( T \log T)$ | State-aware, pre-trained predictor |
>
> ---
>
> We will incorporate this analysis into the appendix of the paper. We thank the reviewer again for the constructive feedback, which has helped enhance the overall quality of our work.

---

> > ### Comment · Reviewer_pacc · 2025-08-07
> >
> > I thank the authors for their response. I am keeping my score at 4 and increasing my confidence. I recommend including a discussion of the computational complexity in the final version.

---

### Note · Authors · 2025-08-14

We would like to thank the reviewers for their constructive feedback and helpful discussions. We sincerely appreciate the time and effort everyone put into this process.


To summarize, the reviewers have recognized the following strengths of our paper -
- novel and well-motivated idea (reviewer pacc, hm7E, 3vb3); and clear, easy-to-follow writing (reviewer pacc, Nmg7, evb3)
- Solid theoretical guarantees (pacc, evb3) with clear proofs (hm7E)
- Practical utilities and extensive (hm7E) / solid (3vb3) experiments
---
### Summary of Revisions

During the discussion period, we incorporated the following changes into the paper:

*Experiments*
- New baselines: DtACL, ECI-cuttoff, and L-ARC (and added reference SAMOCP)
- An ablation study on validity with a misspecified number of regimes.
- An ablation study comparing two aggregation methods with a large number of regimes.

*Analysis, discussion, and writing improvements*
- Runtime analysis of our algorithm vs. baselines
- Discussion on selecting the number of possible regimes
- Clarification of the multivariate setting
- Corrected notations for transition probability and expectation over error rate

All reviewers have acknowledged that their questions have been addressed, and responded positively. The additional results have made the performance evaluation more comprehensive, and our argument for CPTC's utility is now stronger.

---
### Contribution tl;dr

We found a good way to explain the contribution of our work during the rebuttal. We include the paragraph here for reference.

The question CPTC tries to answer is - what happens (to online conformal prediction) when we can *anticipate* distribution shift?

CPTC is designed such that it greatly improves coverage compared to purely reactive methods, while still maintaining the ability to adapt to unknown shifts (and does not need to re-train a model every time step). To our knowledge, there are no other algorithm that explore the same problem; CPTC provides practitioners with a simple and sound solution, when they have the ability (such as via a state space model) to predict shifts.

---

Thank you all again!

---

### Decision · Program_Chairs · 2025-09-17

**Decision:**

Accept (poster)

**Comment:**

This paper introduces a new conformal prediction algorithm (CPTC) for time series with change points, leveraging switching dynamical systems to anticipate distribution shifts. The reviewers generally found the idea novel and well-motivated, with solid theoretical guarantees and clear presentation. While some raised concerns about the scope of experiments and clarifications, the authors have addressed these during the rebuttal with additional baselines, runtime analysis, and ablation studies. Overall, the contributions are timely and relevant, and the paper is a worthwhile addition to the literature.